# A dual role of EZH2 in regulating A-to-I RNA editing and mRNA stability through ADAR

Yang Yi [1,2,20], Yanqiang Li [3,4,5,20], Rui Wang[1], Xufen Yu [6], Qi Liu [1], Chaehyun Yum [1], Yang Zhang [7], Yuanyuan Qiao [8,9,10], Aileen Szczepanski [11], Siqi Wu[1], Qiaqia Li[1], Ladan Fazli[12,13], Jiangchuan Shen[14], Xin Wang [3,4,5], Xiaoling Li [15], Ping Mu [15], Edward M. Schaeffer [1], Heather A. Hundley [16], Hengyao Niu [14], Arul M. Chinnaiyan [8,9,10,17,18], Lu Wang [11], Jinjun Shi [7], Jian Jin [6], Xuesen Dong [12,13], Wei Zhao [19], Kaifu Chen [3,4,5] ✉ & Qi Cao [1,2] ✉

Adenosine-to-inosine (A-to-I) RNA editing, catalyzed by adenosine deaminases acting on RNA (ADARs), is a widespread modification in metazoans. Cumulative evidence has revealed the altered A-to-I editing profiles in cancers, but the underlying mechanism remains unclear. Here, we discover the well-known histone lysine methyltransferase enhancer of zeste homologue 2 (EZH2) as an unexplored ADAR interactor and editing regulator in prostate cancer (PCa). Through competing with interleukin enhancer binding factor 2 (ILF2) for ADAR1 binding, EZH2 reshapes the substrate selectivity of ADAR1 and thus exhibits a bidirectional role in editing regulation. Moreover, EZH2 depletion induces the translational repression of transportin-1 (TRN1), which further results in the accumulation of cytoplasmic ADAR1p110 isoform to protect many oncogenic transcripts from degradation. Consistently, depletion of ADAR1 dramatically enhances the sensitivity of cancer cells and tumors to EZH2 selective degraders. Collectively, our study sheds new light on a link between two layers of epigenetic regulations at histone modification and RNA editing levels, demonstrates a previously uncharacterized role of EZH2 in RNA editing and mRNA stability independently of its lysine methyltransferase activity, and reveals the significance of EZH2-ADAR1 cascade in governing RNA editing and mRNA stability, which may provide additional perspectives for the advancement of EZH2-targeting cancer therapies.

Adenosine deaminases acting on RNA (ADARs) are enzymes that catalyze the adenosine-to-inosine (A-to-I) RNA editing in double-stranded RNA (dsRNA) substrates[1–3]. To date, millions of A-to-I editing sites have been identified across the human transcriptome[4]. Since inosine (I) is considered as guanosine (G) by the cellular machinery, these editing events have the potential to generate diverse biological consequences such as amino acid substitution,

splicing pattern modification and change of noncoding RNA regulations[5]. In humans, ADAR1 (ADAR) and ADAR2 (ADARB1) are the only two catalytically active ADAR family members. Unlike ADAR2 which is relatively abundant in the central nervous system (CNS), ADAR1 is widely expressed throughout the body and contribute to the transcriptomic diversity of many cell types[6]. Notably, aside from RNA editing, ADAR1 further regulates microRNA

---

(miRNA) biogenesis[7,8] and mRNA stability[9,10] independent of A-to-I enzymatic activity.

Recent genomic studies have revealed the aberrant A-to-I editing patterns in a broad range of tumor types including prostate cancer (PCa), which could be largely attributed to the dysregulation of ADAR1 in tumors[11–14]. However, how ADAR1 and A-to-I editing contribute to PCa progression remains controversial. For instance, multiple previous works found that hypo-editing events are more frequent than hyper-editing events in PCa, indicating a repressed ADAR1 activity[11,15,16]. In contrast, it has been reported that the long noncoding RNA (lncRNA) prostate cancer antigen 3 (PCA3) could form a duplex with pre-mRNA of tumor suppressor PRUNE2 in PCa, which subsequently down-regulates PRUNE2 through ADAR1-mediated RNA editing[17]. In this case, elevated ADAR1 expression may benefit PCa development through elimination of PRUNE2. Furthermore, several missense mutations have been identified in androgen receptor (AR) codons, which are intro-duced by RNA editing mechanisms[18]. Interestingly, inhibition of either ADAR1 or ADAR2 had minimal effect on the editing level of AR tran-scripts, implicating the existence of an alternative pathway to mediate AR nucleotide transitions in PCa cells. As such, it is of great importance to decipher the link between ADAR1 activity and altered A-to-I editing landscape in PCa and uncover the molecular basis by which ADAR1 is modulated to provide selective advantages for PCa tumorigenesis.

As the catalytic subunit of polycomb repressive complex 2 (PRC2), EZH2 canonically directs deposition of histone H3 lysine 27 trimethyl-ation (H3K27me3) to maintain a repressive chromatin state[19,20]. Overabundance of EZH2 is closely correlated with aggressiveness and poor prognosis of solid tumors, making it an attractive therapeutic target[21,22]. However, regardless of the massive reactivation of PRC2-repressed tumor suppressor genes, abolishing the enzymatic activity of EZH2 alone has shown limited efficacy in treating EZH2-dependent malignancies such as PCa[23,24]. Meanwhile, compelling evidence including ours has proved that EZH2 could exert multifaceted tumorigenic functions beyond H3K27me3 and PRC2[25–28].

In the present study, we disclose a dual mechanism whereby EZH2 remodels both the A-to-I editome and mRNA turnover landscape in PCa via regulating both editing-dependent and -independent functions of ADAR1. On these bases, we aim to delineate how the EZH2-ADAR1 cascade influences tumorigenesis and cancer therapy.

## Results

### EZH2 directly interacts with ADARs in the nucleus

To identify previously uncharacterized EZH2 binding partners in PCa, we queried our previous co-immunoprecipitation/mass spectrometry (co-IP/MS) data[29] and the constitutively expressed p110 isoform of ADAR1 (ADAR1p110) appeared as a potential EZH2-bound protein. To confirm this observation, we performed co-IP assay using a PCa cell line C4-2. In accordance with the co-IP/MS results, IP of EZH2 pulled down ADAR1p110 and the other two PRC2 core components, EED and SUZ12 (Fig. 1a), while only EZH2 could be detected in the reciprocal ADAR1 IP products (Fig. 1b). This interaction was further validated by IP-IB analysis using LuCaP 35CR[30], a castration-resistant PCa (CRPC) patient-derived xenograft (PDX) model (Fig. 1c, d). Both EZH2 and ADAR1p110 proteins are known to be enriched in the nucleus. To capture the in situ EZH2-ADAR1 interactions, proximity ligation assay (PLA) was next conducted in C4-2 cells. As anticipated, PLA dots representing the endogenous EZH2-ADAR1 interaction were accumu-lated in the nuclei of control cells, but almost completely abolished upon EZH2 or ADAR1 knockdown (Fig. 1e). Although ADAR2 was not listed as an interacting partner of EZH2 by co-IP/MS, the structural and functional similarities of two ADARs prompted us to test the potential of EZH2-ADAR2 interaction. Since ADAR2 expression is too low and hard to detect in PCa model, we ectopically overexpressed Flag-tagged ADAR2 in C4-2 cells, followed by co-IP assay using anti-Flag antibody. As predicted, EZH2, but not EED or SUZ12, was specifically pulled down

by exogenous ADAR2 (Supplementary Fig. 1a). Though both EZH2 and ADARs bear critical RNA-binding capacities[31], the two EZH2-ADAR interactions were well preserved following Ribonuclease A (RNase A) treatment, implying that EZH2 does not rely on RNA for ADAR binding (Fig. 1f and Supplementary Fig. 1b). To further exclude the involvement of other factors, such as DNA or histone, in the EZH2-ADAR interac-tions, purified recombinant proteins of GST-tagged EZH2 and Flag-tagged ADAR1/2 were mixed and subjected to GST pull down assay. In this context, EZH2 proteins were still able to pull down ADAR1/2 pro-teins (Fig. 1g and Supplementary Fig. 1c), confirming that EZH2 and ADAR1/2 could directly interact with each other. In addition, these findings were consolidated by AlphaLISA assays using purified EZH2 and ADAR1/2 proteins, which revealed a strong binding affinity for both interactions (Fig. 1h and Supplementary Fig. 1d–f).

EZH2 mainly comprises two SANT protein-protein interaction domains, one cysteine-rich CXC domain and an enzymatic SET domain (Fig. 1i). ADAR1p110 contains a single Z-DNA binding domain, three dsRNA binding domains (dsRBDs) and a deaminase domain in the C-terminal (Fig. 1j), while ADAR2 consists of two dsRBDs followed by one deaminase domain (Supplementary Fig. 1g). To uncover the interacting domains that are required for the EZH2-ADAR interactions, we co-transfected serially truncated mutants of Myc-tagged EZH2 or Flag-tagged ADAR1/2 constructs with their full-length partner plasmids into HEK293T cells for co-IP analyses. As revealed in Fig. 1k, except for EZH2ΔSANT2, all other truncated mutants and full-length of EZH2 pulled down full-length ADAR1. Meanwhile, only ADAR1ΔdsRBD123 failed to pull down full-length EZH2 (Fig. 1l). In terms of ADAR2, EZH2ΔSANT2 was also characterized as the only EZH2 mutant which was unable to pull down full-length ADAR2 (Supplementary Fig. 1h), while full-length EZH2 could be detected in all ADAR2 IP products excluding ADAR2ΔdsRBD12 (Supplementary Fig. 1i). Collectively, these findings suggest that all dsRBDs of both ADAR proteins retain the EZH2-binding capacity through EZH2's SANT2 domain.

### EZH2 reshapes the A-to-I editome in PCa cells primarily through ADAR1

The aforementioned interactions led us to investigate whether EZH2 could participate in the RNA editing regulation. To this end, we per-turbed the expression of EZH2 as well as two ADARs in C4-2 cells by siRNAs, followed by RNA-seq to map the editing changes. To avoid the biases introduced by single nucleotide polymorphisms (SNPs) or somatic mutations, we further sequenced the entire genome of C4-2 cells as reference. The A-to-G (I) conversion was ranked as the most frequent type of nucleotide changes, accounting for 73.6% of the changes observed across the transcriptome of C4-2 cells (Fig. 2a). Notably, although EZH2 deficiency did not appear to have any noticeable impact on ADAR expression, the overall A-to-I editing ratio was significantly reduced in EZH2-deficient C4-2 cells when compared to the control group (Fig. 2b and Supplementary Fig. 2a). As controls, a dramatic downregulation of global editing was also found upon ADAR1 depletion, while further knockdown of already lowly expressed ADAR2 only led to a slight editing decrease in C4-2 cells (Fig. 2b). Similar results were also found when using PCa patient-derived organoids (PDOs) MDA-PCa-174 (Supplementary Fig. 2b, c). To consolidate this observation, we first tested a series of public EZH2-knockdown RNA-seq datasets in PCa and other cancer cell lines to reconfirm the decreased editing pattern (Supplementary Fig. 2d). Moreover, after dividing the Stand Up To Cancer (SU2C) metastatic PCa patient sam-ples based on EZH2 expression[32], the editing level in EZH2-high group was statistically higher than the EZH2-low group without any change in terms of ADAR1 expression (Supplementary Fig. 2e). Interestingly, treatment of C4-2 cells with EZH2 enzymatic inhibitors of GSK126 and EPZ6438 barely affected the global editing level (Supplementary Fig. 2f), indicating that EZH2 mainly regulates A-to-I editing in a way independently of its methyltransferase activity.

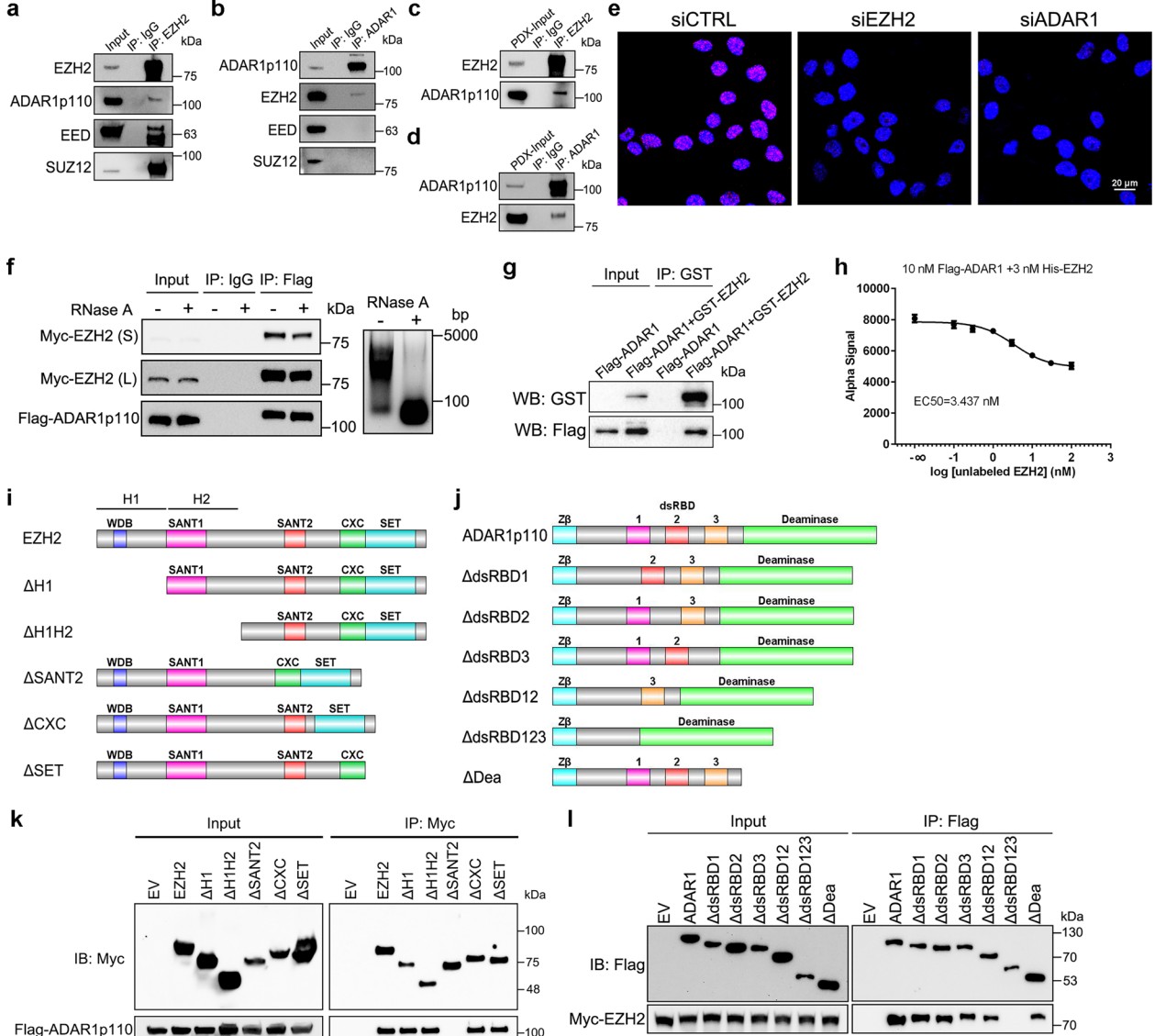

**Fig. 1 | Direct interaction between EZH2 and two ADARs.** C4-2 cells were lysed and subjected to co-IP assay using anti-EZH2 (**a**) or anti-ADAR1 (**b**) antibody, followed by western blot with indicated antibodies. Rabbit IgG was used as negative control. The LuCaP 35CR PDX tissues were lysed and subjected to co-IP assay using anti-EZH2 (**c**) or anti-ADAR1 (**d**) antibody, followed by western blot with indicated antibodies. Rabbit IgG was used as negative control. **e** Control, EZH2-deficient or ADAR1-deficient C4-2 cells were subjected to PLA with anti-EZH2 and anti-ADAR1 antibodies. PLA signals were visualized as red foci and nuclei by DAPI staining. Scale bar: 20 μm. **f** C4-2 cells co-transfected with Myc-tagged EZH2 and Flag-tagged ADAR1p110 were lysed and subjected to co-IP assay using anti-Flag antibody. Before IP, cell lysates were treated with or without RNase A. Both the shorter (S) and longer (L) exposures of EZH2 bands were presented. Agarose gel demonstrated the complete digestion of total RNA. **g** Purified proteins of GST-tagged EZH2 and Flag-tagged ADAR1p110 were subjected to GST pull down assay, followed by western blot with indicated antibodies. **h** Inhibition of His-tagged EZH2 and Flag-tagged ADAR1p110 binding by unlabeled EZH2 in AlphaLISA displacement assay. Data represent Mean ± SD for $n = 3$ biologically independent experiments. **i** Schematic diagrams of EZH2 protein and its truncation mutants. The homology domain 1 (H1) contains WDB domain, while the homology domain 2 (H2) contains the first SANT domain. **j** Schematic diagrams of ADAR1p110 protein and its truncation mutants. Zβ represents the Z-DNA binding domain β. **k** Co-IP of Flag-tagged ADAR1p110 with full-length or truncation mutants of Myc-tagged EZH2, followed by western blot with indicated antibodies. **l** Co-IP of Myc-tagged EZH2 with full-length or truncation mutants of Flag-tagged ADAR1p110, followed by western blot with indicated antibodies. Source data are provided as a Source Data file.

With respect to the individual editing event, the majority of EZH2-affected A-to-I editing sites reside in the three prime untranslated region (3′-UTR), followed by intergenic and intronic regions (Fig. 2c). Intriguingly, both over-edited (editing ratio increased) and under-edited (editing ratio decreased) sites ($N_{over} = 244$ vs $N_{under} = 548$) could be identified in EZH2-deficient C4-2 cells (Fig. 2d and Supplementary Data 1), indicating an opposing effect of EZH2 in controlling A-to-I editing. In parallel, most sites altered upon ADAR1 depletion were under-edited ($n_{(over)} = 69$ vs $n_{(under)} = 4326$), while the number of over-edited sites was inversely higher than the under-edited sites in ADAR2-

deficient cells ($n_{(over)} = 502$ vs $n_{(under)} = 312$) (Fig. 2d and Supplementary Data 1). We then defined the sites that showed significantly decreased editing frequency upon ADAR1/2 knockdown as the ADAR1/2-mediated editing sites to compare with our identified EZH2-affected sites. As revealed in Fig. 2e, f, 70.4% ($n = 386$) of under-edited and 20.5% ($n = 50$) of over-edited sites upon EZH2 deficiency could be classified as the ADAR1-mediated editing sites, while only 17.5% ($n = 96$) of under-edited and one single over-edited site upon EZH2 deficiency were overlapped with the ADAR2-mediated editing sites. Moreover, among the 548 EZH2-regulated under-edited sites, 80 sites are

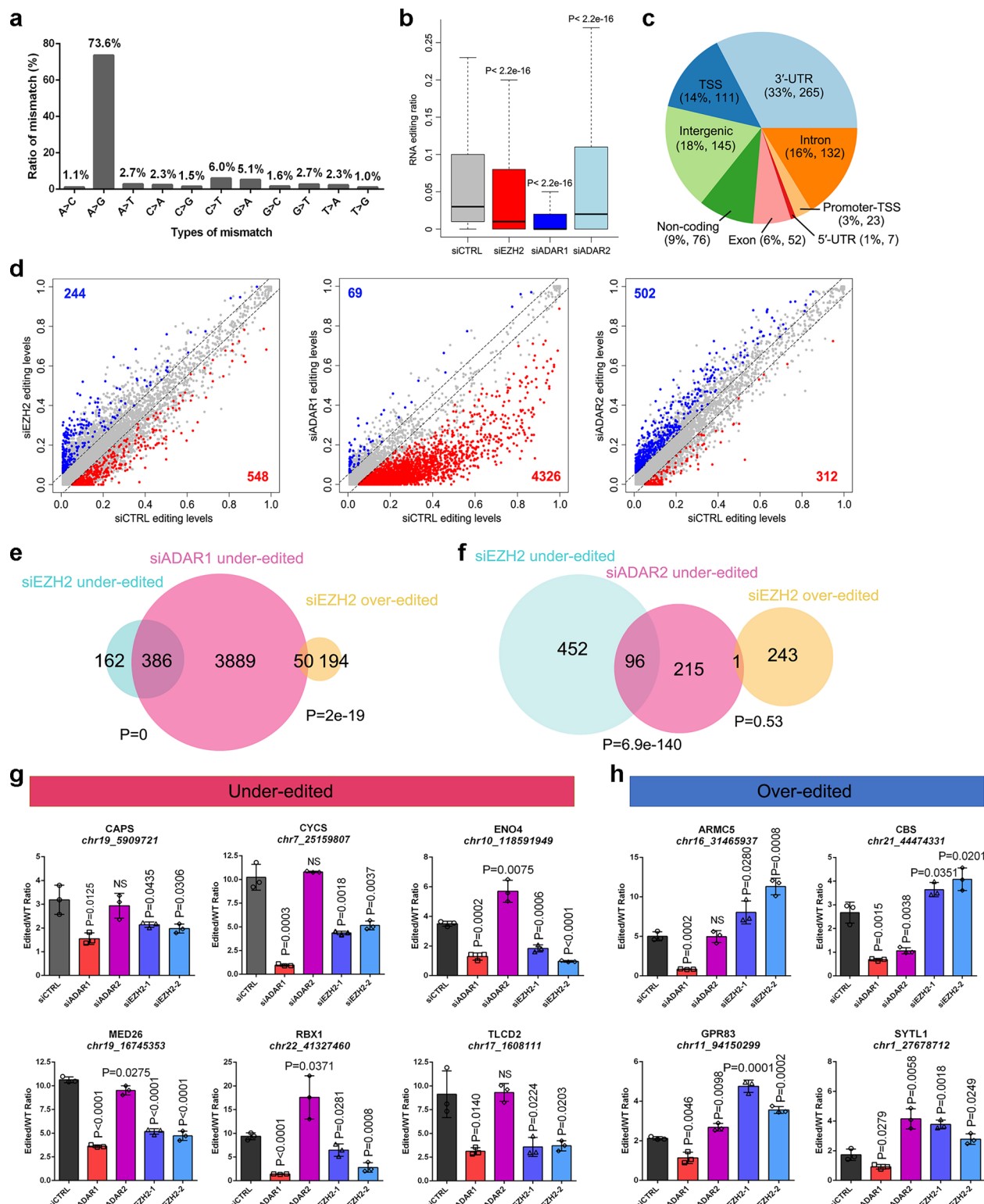

**Fig. 2 | EZH2 functions as a bidirectional editing regulator in PCa cells.**
**a** Distribution of 11 types of nucleotide changes across the entire transcriptome of C4-2 cells, as revealed by RNA-seq. **b** Box plot showing the global A-to-I editing ratio in each group as indicated. *P* values were calculated by two-tailed unpaired Wilcoxon's test. $n = 4$ for siCTRL, $n = 4$ for siEZH2, $n = 6$ for siADAR1, and $n = 5$ for siADAR2. Box plots show the median (center line), the interquartile range (box, 25th–75th percentiles), and whiskers indicating the minimum and maximum values. **c** Pie chart showing the distribution of EZH2-affected A-to-I editing sites over annotated genomic regions of C4-2 cells. TSS, transcription start site. **d** Scatter plots of comparison of individual editing sites (dots) between control and

knockdown RNA-seq of EZH2, ADAR1 and ADAR2 in C4-2 cells. Venn diagrams showing the overlap between ADAR1- (**e**) or ADAR2- (**f**) mediated editing sites and EZH2-affected editing sites. P values were calculated by one-tailed Fisher's exact test. Representative RESSq-PCR results to validate the relative editing changes of under-edited (**g**) and over-edited (**h**) sites in C4-2 cells upon EZH2 deficiency. The edited/WT ratio reflects relative differences in editing efficiency between conditions, not absolute editing percentages. *P* values were calculated using two-sided student's t-test. Data represent Mean ± SD from $n = 3$ biologically independent experiments. Source data are provided as a Source Data file.

co-regulated by all three factors, while only 16 sites are shared between EZH2 and ADAR2 without ADAR1 involvement (Supplementary Fig. 2g). In addition, when all the ADAR1/2-shared editing sites were removed, we could still observe a highly significant overlap between the ADAR1-regulated sites and EZH2-regulated sites (Supplementary Fig. 2h). In contrast, only a comparatively weaker but still statistically significant overlap could be found between EZH2 and ADAR2-regulated sites (Supplementary Fig. 2i). All these data proved that EZH2 mainly modulates A-to-I editing in PCa cells through ADAR1, while also exerting a modest but measurable influence via ADAR2.

Next, to test the accuracy of our sequencing results, 30 EZH2-affected sites (20 under-edited and 10 over-edited) were randomly selected for validation using an RNA editing fingerprint approach of RNA editing site-specific quantitative PCR (RESSq-PCR)[33]. As a result, 80% (16) under-edited sites and 80% (8) over-edited sites were successfully validated to be regulated by EZH2 in C4-2 cells (Fig. 2g, h and Supplementary Data 2). Consistently, most of the tested EZH2-affected sites also underwent a drastic under-editing upon ADAR1 knockdown but remained unchanged or even over-edited upon ADAR2 depletion. Moreover, to confirm that these EZH2-affected editing changes are not restricted in a certain prostate cell type, we overexpressed EZH2 in human primary prostate epithelial cells (PrEC) which bear rare endogenous EZH2 expression (Supplementary Fig. 2j). RESSq-PCR results showed that the majority of our tested sites could also been validated in EZH2-overexpressing PrEC model, in which circumstance the editing ratios were changed towards an opposite direction as occurred in EZH2-knockdown C4-2 cells (Supplementary Fig. 2k, l and Supplementary Data 2). In summary, our data suggested that, mainly through ADAR1, EZH2 reshapes the A-to-I editing profiles of PCa cells in a bidirectional manner.

### Identification of *chr12: 69237519* at *MDM2* transcript as a functional editing site affected by EZH2 in PCa

To gain a deeper insight into the EZH2-remodeled RNA editing in PCa, we decided to perform a comprehensive investigation on a specific editing site (*chr12: 69237519*) at the 3′-UTR of *MDM2* transcript, which was significantly under-edited upon EZH2/ADAR1 depletion (Supplementary Data 1). By utilizing Sanger sequencing and RESSq-PCR assays, we first validated this site in both C4-2 and PrEC models (Fig. 3a, b, Supplementary Fig. 3a, b). Strikingly, conjoint analyses of the Cancer Genome Atlas (TCGA) and Genotype-Tissue Expression (GTEx) databases observed a higher editing frequency of *chr12: 69237519* in PCa tissues relative to normal prostate controls, which was accompanied by the overexpression of EZH2, but not ADAR1 (Fig. 3c). Moreover, instead of ADAR1, expression of EZH2 was positively correlated with editing level of this site in PCa (Fig. 3d). Indeed, localized PCa is not featured by abnormal ADAR1 expression, as immunohistochemistry (IHC) staining of ADAR1 proteins in a series of PCa specimens observed no significant difference between tumor and the adjacent benign tissues (Supplementary Fig. 3c, d). Therefore, other than ADAR1, elevated expression of EZH2 is more likely to be responsible for the increased editing level of *chr12: 69237519* in PCa.

Of note, this specific site is harbored in the seeding sequences of miR-204 and −211 (Fig. 3e, the upper panel). Consistently, endogenous MDM2 expression was markedly decreased in C4-2 cells transduced with mimics of both miRNAs (Supplementary Fig. 3e), confirming their regulatory connections. To examine whether the A-to-I (G) conversion at *chr12: 69237519* could affect the binding of two miRNAs to *MDM2* transcript, dual-luciferase reporters with either wildtype (WT, unedited) or edited MDM2 3′-UTR were introduced for investigation (Fig. 3e, the lower pane l). After overexpression of miR-204 or −211 mimics, both WT and edited reporters showed a decreased reporter activity (Fig. 3f). Remarkably, the reduction was much more evident in WT group as compared to the edited group, suggesting that RNA editing of *chr12: 69237519* enables MDM2 to partially evade targeting

by miR-204 and −211. In keeping with our hypothesis that EZH2 maintains the over-editing of *chr12: 69237519*, EZH2 suppression in C4-2 cells led to the downregulation of MDM2 without affecting the expression of both miRNAs (Fig. 3g and Supplementary Fig. 3f). In addition, EZH2 depletion failed to further decrease MDM2 expression in ADAR1-deficient cells (Supplementary Fig. 3g), supporting a notion that EZH2 promotes MDM2 expression through modulation of ADAR1-mediated RNA editing. Taking *chr12: 69237519* as an example, we next investigated whether the methyltransferase activity is required for EZH2 to regulate RNA editing. To achieve this, we overexpressed various RNAi-resistant EZH2 mutants in EZH2-deficient C4-2 cells, followed by measurement of *chr12: 69237519* editing ratio and MDM2 expression in each group. As shown in Fig. 3h and Supplementary Fig. 3h, re-expression of either WT or catalytically dead H689A-mutant EZH2, but not EZH2ΔSANT2, could fully rescue the editing of *chr12: 69237519*. Meanwhile, the protein level of MDM2 was coordinately changed (Fig. 3i). Together, these observations prove that interaction with ADAR1 is essential for EZH2 to mediate RNA editing of *MDM2* transcript in a methylation-independent manner.

### Selective binding to EZH2 or ILF2 determines the editing substrates of ADAR1

We next sought to discover the mechanism underlying EZH2-mediated site-specific editing regulation. In accordance with our RNA-seq data, western blot analysis revealed no alteration in ADAR1p110 expression upon EZH2 depletion (Supplementary Fig. 4a). Similarly, neither EZH2 expression nor H3K27me3 level was changed upon ADAR1 suppression (Supplementary Fig. 4b). Since EZH2 is capable of methylating proteins other than histone[34], we next wondered whether ADAR1p110 is a distinct EZH2 substrate. ADAR1p110 proteins from control or EZH2-deficient C4-2 cells were immunoprecipitated and subjected to western blot to measure the lysine methylation levels. As depicted in Supplementary Fig. 4c, both mono-/di- and tri-methylation levels of ADAR1p110 were barely changed after EZH2 depletion, suggesting that EZH2 is unlikely to methylate lysine residues of ADAR1p110. Homodimerization of ADARs is critical for their enzymatic activities[35]. However, co-IP results showed that EZH2 knockdown did not affect the abundance of Myc-tagged ADAR1p110 coimmunoprecipitated with Flag-tagged ADAR1p110 (Supplementary Fig. 4d), excluding the possibility that EZH2 disturbs the formation of ADAR1 homodimer.

We then focused on the change of ADAR1 interactome since the preference of ADAR1 to interact with other RNA-binding proteins (RBPs) could dictate its editing substrate specificity[36]. By conducting the quantitative co-IP/MS assay in C4-2 cells, we observed that, more interleukin enhancer binding factor 2 (ILF2), a previously reported editing regulator[36], bound to ADAR1p110 in EZH2-knockdown groups (Fig. 4a). To validate this proteomics result, we pulled down endogenous ADAR1p110 along with its interacting partners from control or EZH2-deficient C4-2 cells, followed by western blot to compare the abundance. Whereas EZH2 knockdown had no impact on ILF2 expression, much more ILF2 proteins could be coimmunoprecipitated by ADAR1p110 in EZH2-deficient cells than those in control cells (Fig. 4b). As compared, ILF3, a related protein of ILF2 which also functions in editing regulation[36], showed no change in ADAR1-binding capability upon EZH2 deficiency. We then hypothesized that EZH2 competes with ILF2 to interact with ADAR1. To validate this, we first confirmed that ILF2 binds to the first dsRBD (dsRBD1) of ADAR1p110 (Fig. 4c). Moreover, AlphaLISA assay revealed a gradually reduced EZH2-ADAR1 interaction signals as the concentration of ILF2 proteins increased (Fig. 4d and Supplementary Fig. 4e), indicating that the binding of ADAR1p110 with EZH2 or ILF2 is mutually exclusive. To consolidate this finding, we pulled down endogenous ADAR1p110 along with its interacting EZH2 proteins in the context of ILF2 deficiency. Interestingly, although ILF2 inhibition led to an unexpected

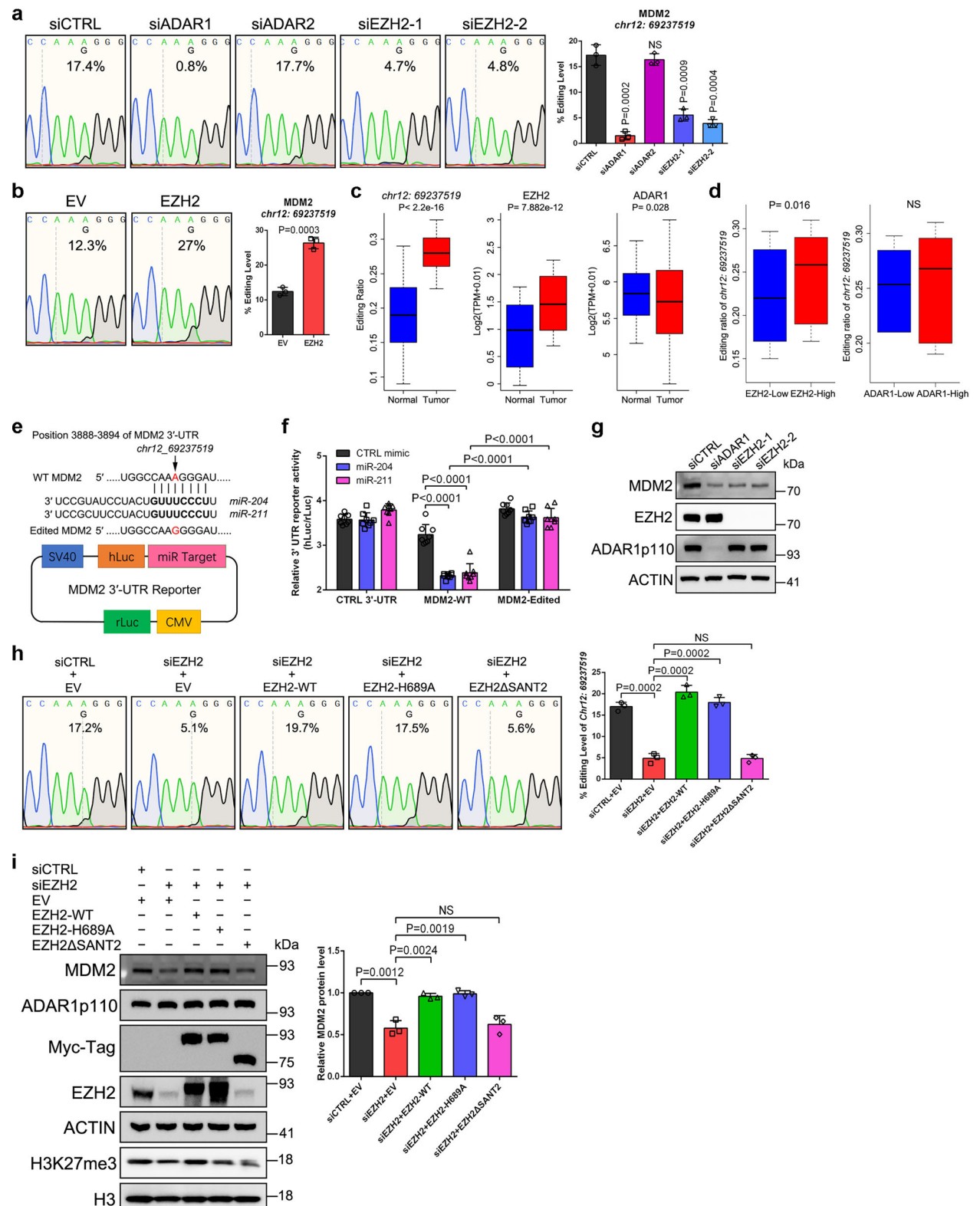

EZH2 downregulation, more EZH2 proteins could still be coimmunoprecipitated by ADAR1p110 in ILF2-knockdown groups (Supplementary Fig. 4f). Consistently, more PLA dots representing the EZH2-ADAR1 interaction were also captured in ILF2-deficient C4-2 cells (Supplementary Fig. 4g). Size-exclusion chromatography analyses were then carried out using nuclear extracts from control and EZH2-deficient C4-2 cells. While the elution profile of ADAR1p110 remained invariable upon EZH2 knockdown, EZH2 suppression led to an obvious shift of ILF2 elution profile towards ADAR1p110 fractions (Fig. 4e), reflecting a stronger ILF2-ADAR1 interaction.

To deeply dissect the consequence of competitive ADAR1 binding between EZH2 and ILF2, we performed the enhanced crosslinking and immunoprecipitation-sequencing (eCLIP-seq) in C4-2 cells to define the RNA targets of EZH2, ADAR1 and ILF2, respectively (Supplementary Data 3). As a result, the group of RNAs that are bound by each RBP was significantly overlapped with each other (Fig. 4f). After integration with

**Fig. 3 | EZH2 promotes MDM2 expression through hyper-editing of *chr12: 69237519*.** Sanger sequencing chromatograms illustrating editing frequency of *chr12: 69237519* in C4-2 cells undergoing ADAR or EZH2 knockdown (**a**) or PrEC undergoing EZH2 overexpression (**b**). Graph showing the quantification of editing level in each group. Data represent Mean ± SD from I = 3 biologically independent experiments. **c** The editing ratio of *chr12: 69237519* and the mRNA levels of EZH2 and ADAR1 were compared between normal (GTEx, n = 152) and PCa tumor (TCGA, n = 495) samples. *P* values were calculated by two-tailed unpaired Wilcoxon's test. Box plots show the median (center line), the interquartile range (box, 25th–75th percentiles), and whiskers indicating the minimum and maximum values. **d** Box plot showing the editing ratio of *chr12: 69237519* in prostate specimens with different EZH2 or ADAR1 levels (n = 50 independent samples per group) using data from GTEx and TCGA. *P* values were calculated by two-tailed unpaired Wilcoxon's test. Box plot format as in (**c**). **e** Structure of the WT and edited MDM2 3′-UTR reporters. The mutation site with an A-to-G (I) conversion was shown in the upper panel. **f** Each of the indicated reporter plasmids was transfected into C4-2 cells overexpressing control, miR-204 or miR-211 mimics, followed by detection of luciferase activities through dual-luciferase assay. Data represent Mean ± SD from n = 4 biologically independent experiments. **g** Western blot to detect the change of MDM2 protein level upon ADAR1 or EZH2 suppression in C4-2 cells. **h** Sanger sequencing chromatograms showing the rescue effects of a series of EZH2 mutants on editing ratio of *chr12: 69237519* affected by EZH2 depletion. Graph showing the quantification of editing level. Data represent Mean ± SD from n = 3 biologically independent experiments. **i** Western blot to detect the change of MDM2 protein level upon forced expression of various EZH2 mutants in EZH2-deficient C4-2 cells. The graph represents the relative MDM2 protein level. Data represent the mean ± SD from n = 3 biologically independent measurements. Statistical significance was assessed using two-sided student's *t* test unless otherwise stated. Source data are provided as a Source Data file.

our aforementioned RNA-seq results, all three RBPs showed evidence of binding at the A-to-I editing sites (Fig. 4g). Remarkably, the number of ILF2-bound RNAs undergoing over-editing upon EZH2 knockdown was much higher than expected (Fig. 4h), which accounts for 61% of the RNAs with EZH2 deficiency-induced over-edited sites (Fig. 4i). In comparison, the number of EZH2-bound RNAs undergoing under-editing upon EZH2 knockdown was also higher than expected (Fig. 4j), which accounts for 59% of the RNAs with EZH2 deficiency-induced under-edited sites (Fig. 4k). Further analysis suggested that, as compared with ILF2, EZH2-binding regions preferentially overlap with the under-edited sites upon EZH2 knockdown. In contrast, the ILF2-binding regions show greater overlap with the over-edited sites following EZH2 depletion (Supplementary Fig. 4h). All these findings suggested that the strengthened ADAR1-ILF2 interaction upon EZH2 suppression may change the editing substrate selectivity of ADAR1, which subsequently results in the bidirectional alteration of editing in EZH2-deficient cells. Therefore, for those EZH2-bound RNAs without ILF2-binding capacity (e.g., *MDM2*, as depicted in Fig. 4l), their editing sites are prone to be under-edited upon EZH2 deficiency (e.g., *chr12: 69237519* of *MDM2*, as shown in Fig. 3). Meanwhile, for those ILF2-bound RNAs lacking EZH2-binding capacity (e.g., *SYTL1*, as depicted in Supplementary Fig. 4i), their editing sites are more likely to be over-edited upon EZH2 depletion (e.g., *chr1_27678712* of *SYTL1*, as presented in Fig. 2h).

## EZH2 suppression induces cytoplasmic translocation of ADAR1p110 through translational downregulation of TRN1

Since ADAR1p110 mainly edits RNA in the nucleus but further exerts editing-independent functions in the cytoplasm[10,37], we next examined the subcellular distribution of ADAR1p110 under EZH2 depletion. As expected, endogenous ADAR1p110 was found predominantly in the nucleus of control C4-2 cells. However, the cytoplasmic fraction of ADAR1p110 was significantly increased in the absence of EZH2 (Fig. 5a). To further validate this finding, we transfected GFP-tagged ADAR1p110 plasmids into C4-2 cells and visualized by fluorescence microscopy along with western blot analysis. As presented in Fig. 5b and Supplementary Fig. 5a, repression of EZH2 also stimulated the cytoplasmic accumulation of GFP-tagged ADAR1p110, with a phenotype similar to the previously reported stress-activated ADAR1p110 shuttling to the cytoplasm[10].

To unveil the reason for the translocation of ADAR1p110 upon EZH2 suppression, we checked the expression of transportin-1 (TRN1, also known as TNPO1) and exportin-5 (XPO5), which controls the nuclear import and export of ADAR1p110 proteins, respectively[38]. As shown in Fig. 5c, the protein level of TRN1, but not XPO5, was substantially reduced upon EZH2 knockdown. Intriguingly, TRN1 mRNA level was barely altered upon EZH2 suppression, albeit the significant change of protein level (Supplementary Fig. 5b). This finding was reminiscent of our recent report that EZH2 could exert a PRC2-

independent role to facilitate the internal ribosome entry site (IRES)-dependent translation initiation in cancer cells through fibrillarin (FBL)[27]. By checking our ribosome profiling (Ribo-seq) data within this article, we found that TRN1 was classified as a buffering down gene with minor decrease at mRNA level but significant reduction at protein level upon EZH2 inhibition, which matched our current observations. To test whether TRN1 is regulated by EZH2 at translational level, we first confirmed that TRN1 expression was also decreased upon FBL knockdown (Supplementary Fig. 5c). In addition, treatment of C4-2 cells with EZH2 protein inhibitor DZNep which decreases EZH2 protein level, but not enzymatic inhibitors of GSK126 and EPZ6438, led to the downregulation of TRN1 (Fig. 5d), indicating that EZH2 controls TRN1 expression beyond its methyltransferase activity. A potential IRES element has been predicted at the 5′-UTR of TRN1[39]. Next, we cloned this putative IRES sequence into a bicistronic luciferase vector for analysis (Fig. 5e, the upper panel). In line with our hypothesis, TRN1 IRES showed a reduced activity in both FBL- and EZH2-deficient cells (Fig. 5e, the lower panel). To reinforce these observations in vivo, we first searched the TCGA database and confirmed that mRNA level of TRN1 was similar between normal prostates and PCa tissues (Fig. 5f). In sharp comparison, IHC results of PCa tissue microarray (TMA) slide revealed that TRN1 proteins were hardly detected in benign prostate tissues but upregulated significantly with the advancement of PCa (Fig. 5g). Meanwhile, IHC assay was conducted in serially sectioned PCa TMA slides using both anti-EZH2 and anti-TRN1 antibodies. As a result, a strongly positive correlation was found between expressions of EZH2 and TRN1 (Fig. 5h, i), further supporting the EZH2-mediated translational regulation of TRN1.

Finally, we aimed to determine whether diminished TRN1 expression is responsible for the enrichment of cytoplasmic ADAR1p110 upon EZH2 suppression. We first observed that inhibition of TRN1 alone resulted in the dramatic retention of ADAR1p110 in the cytoplasm (Supplementary Fig. 5d). More importantly, ectopic expression of TRN1 in EZH2-deficient cells significantly restored the nuclear localization of ADAR1p110 (Fig. 5j, k). Above all, these data indicate that EZH2-mediated regulation of TRN1 translation has a downstream effect on the subcellular localization of ADAR1p110, which may contribute to the observed reduction of global editing level in EZH2-deficient PCa cells (Fig. 2b and Supplementary Fig. 2b).

## EZH2 influences the dynamic of mRNA stability through cytoplasmic ADAR1p110

Independent of its RNA editing function which is mostly restricted in the nucleus, the cytoplasmic ADAR1p110 competes with staufen-1 (STAU1) to bind to a set of gene transcripts with 3′-UTR dsRNA structures and thus antagonizes the STAU1-mediated mRNA decay[10]. To investigate whether EZH2 depletion could affect the mRNA degradation process through translocation of ADAR1p110, we treated control, ADAR1- and EZH2-deficient C4-2 cells with Actinomycin D (ActD) to

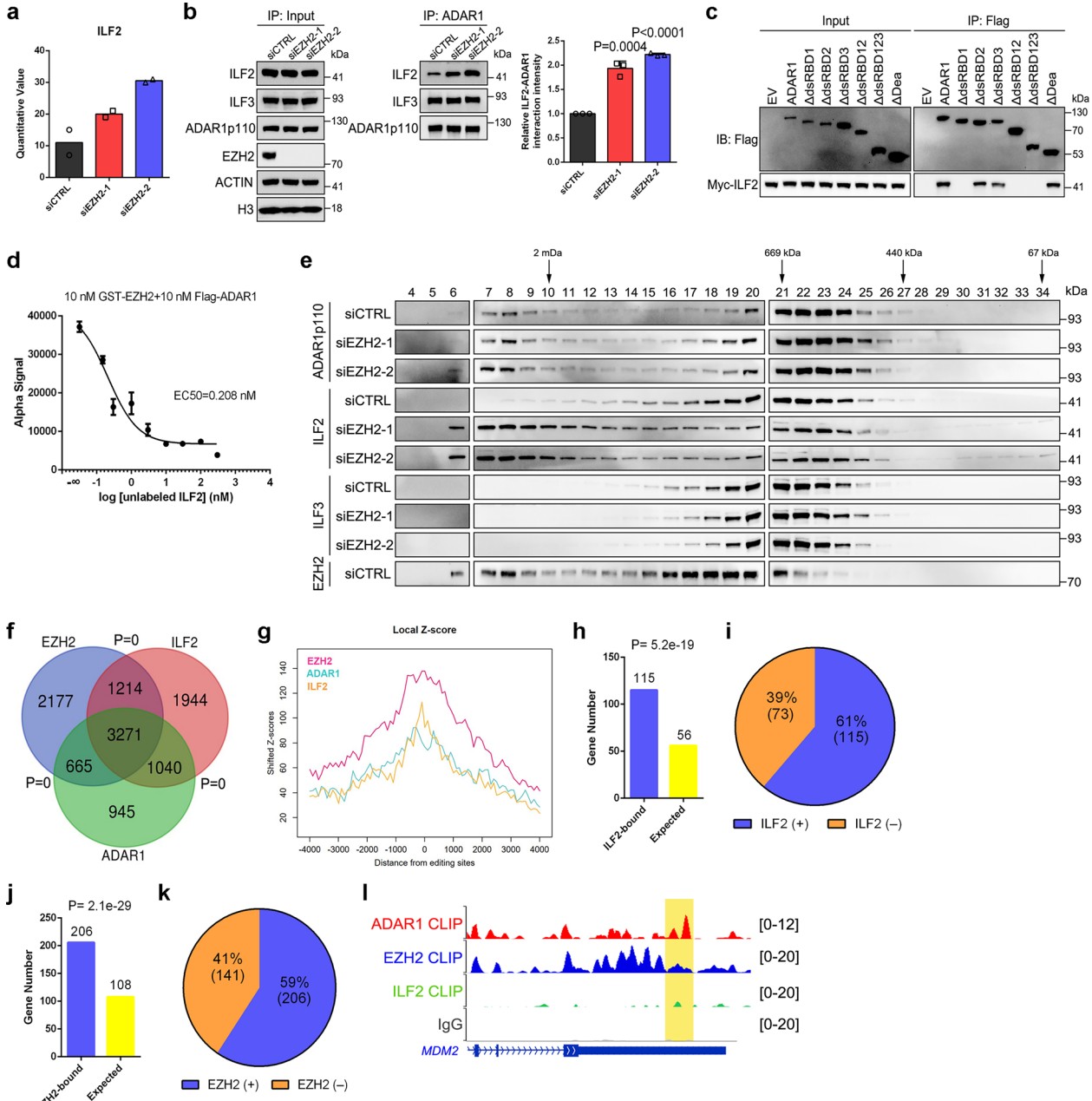

**Fig. 4 | EZH2-mediated remodeling of ADAR1 interactome determines the selectivity of ADAR1 editing substrates. a** Graph showing the quantitative value of ADAR1-binding ILF2 proteins in control or EZH2-knockdown groups, as revealed by quantitative co-IP/MS assay. Data represent Mean ± SD from $n = 2$ biologically independent experiments. **b** Left: Co-IP of ILF2 and ILF3 with ADAR1 in control and EZH2-deficient C4-2 cells, followed by western blot with indicated antibodies. Right: the graph represents the relative ILF2-ADAR1 interaction intensity in each group. Data represent the mean ± SD from $n = 3$ biologically independent measurements. Statistical significance was determined by two-tailed Student's $t$ test. **c** Co-IP of Myc-tagged ILF2 with full-length or truncation mutants of Flag-tagged ADAR1p110, followed by western blot. **d** Inhibition of GST-tagged EZH2 and Flag-tagged ADAR1p110 binding by unlabeled ILF2 in AlphaLISA displacement assay. Data represent Mean ± SD for $n = 3$ biologically independent experiments. **e** Size-exclusion analysis of nuclear extracts from control and EZH2-deficient C4-2 cells; western blot detection of ADAR1p110, ILF2, ILF3, and EZH2 (control). **f** Venn diagram showing the overlap among EZH2-, ADAR1- and ILF2-bound gene transcripts. $P$ values were calculated by one-tailed Fisher's exact test. **g** Z scores showing the observed binding strengths of EZH2, ADAR1 and ILF2 around the A-to-I editing sites, as revealed by eCLIP-seq and RNA-seq results. **h** Graph showing that the number of ILF2-bound RNAs containing at least one over-edited site upon EZH2 knockdown is higher than expected. $P$ values were calculated by one-tailed Fisher's exact test. **i** Pie chart showing the ratio of ILF2-bound RNAs in RNAs that contain at least one over-edited site upon EZH2 knockdown. **j** Graph showing that the number of EZH2-bound RNAs containing at least one under-edited site upon EZH2 knockdown is higher than expected. $P$ values were calculated by one-tailed Fisher's exact test. **k** Pie chart showing the ratio of EZH2-bound RNAs in RNAs that contain at least one under-edited site upon EZH2 knockdown. **l** Representative genome browser tracks to show eCLIP-seq data at the loci of *MDM2*. The region covering the editing site of *chr12: 69237519* was marked. Source data are provided as a Source Data file.

block nascent RNA synthesis, and collected samples at a series of timepoints for RNA-seq (Fig. 6a). As expected, loss of ADAR1 significantly reduced the RNA stability globally, while an overall increased RNA half-life was observed in EZH2-deficient cells (Fig. 6b). Further analysis

revealed that genes with increased RNA half-lives upon EZH2 knockdown were significantly overlapped with genes undergoing decreased RNA half-lives upon ADAR1 knockdown (Fig. 6c). In sharp contrast, no evident overlapping was found between genes with decreased RNA

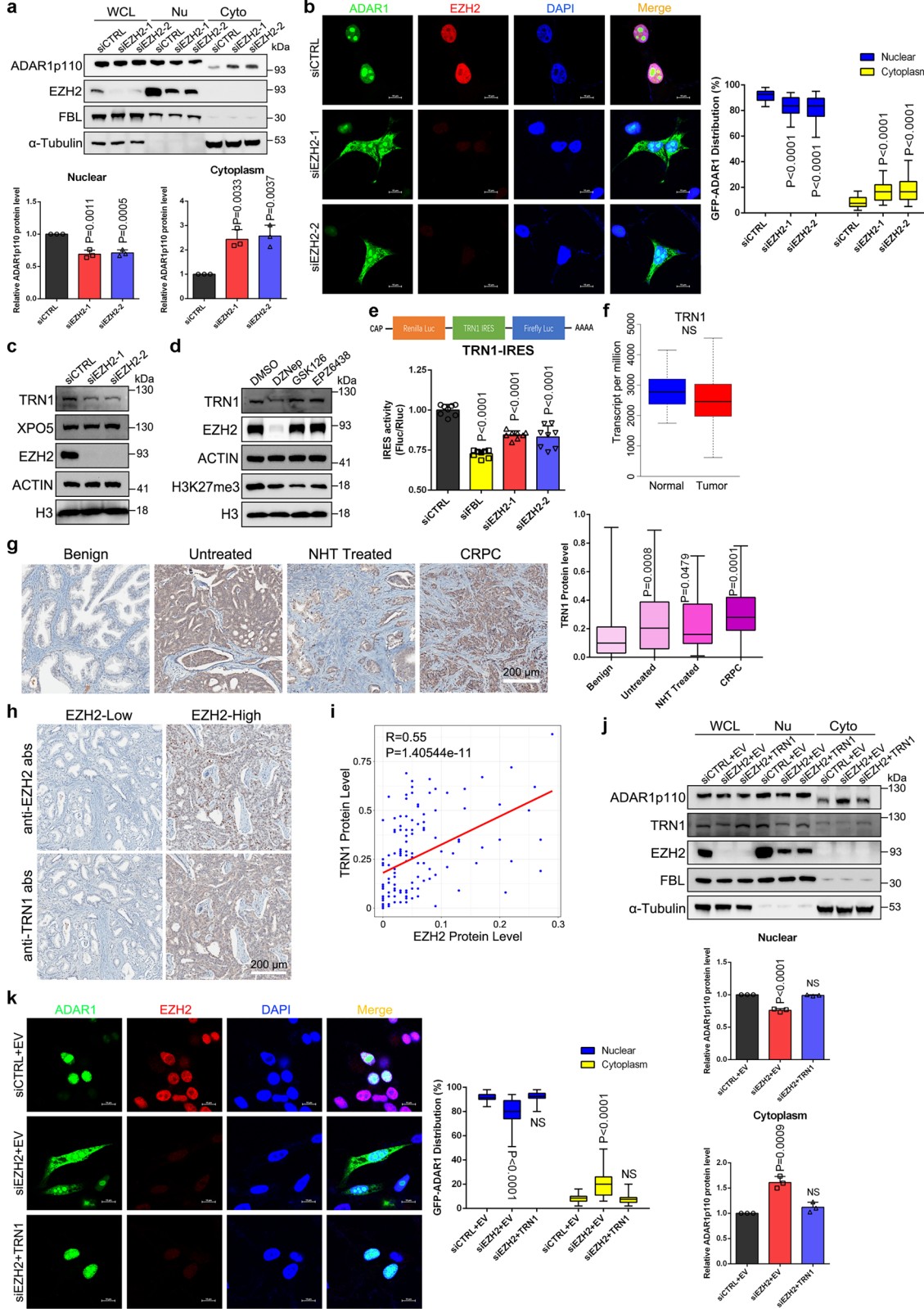

stability upon EZH2 knockdown and those with increased RNA stability upon ADAR1 suppression (Fig. 6d). Thus, these data suggested that the role of EZH2 in modulating RNA stability is correlated with ADAR1.

We next focused on the 3810 gene transcripts with prolonged half-lives upon EZH2 deficiency while weakened stability upon ADAR1 suppression (Fig. 6c and Supplementary Data 4), which are deemed to be regulated by the EZH2-ADAR1 cascade in terms of RNA decay rate.

Unexpectedly, these genes were particularly enriched into a number of cancer-specific oncogenic pathways (Fig. 6e), implying that elimination of EZH2 may not only release the epigenetic silencing of key tumor suppressors at transcription level, but also inversely stabilize a set of oncogenes post-transcriptionally. To further confirm this, we selected four gene transcripts of CCNG1, ATM, YES1 and SMARCD1 for validation. All of these candidates are known ADAR1 binding targets with

**Fig. 5 | EZH2 regulates the nucleocytoplasmic shuttling of ADAR1p110 through TRN1. a** Western blot to detect the distribution of ADAR1p110 proteins in whole cell lysates (WCL), nuclear (Nu) and cytoplasmic (Cyto) fractions upon EZH2 depletion in C4-2 cells. FBL and α-tubulin were used as nuclear and cytoplasmic markers, respectively. Graphs represent the relative ADAR1p110 protein level. Data represent the mean ± SD from $n = 3$ biologically independent measurements. **b** Representative fluorescence images of control or EZH2-deficient C4-2 cells expressing GFP-tagged ADAR1p110. Endogenous EZH2 were co-stained and the nuclei were visualized by DAPI (Scale bar: 10 μm). Graphs quantify nuclear/cytoplasmic ADAR1p110 (Mean ± SD, $n = 30$ cells). Box plots show the median (center line), the interquartile range (box, 25th–75th percentiles), and whiskers indicating the minimum and maximum values. Western blots showing TRN1 and XPO5 levels upon EZH2 knockdown (**c**) or treatment with EZH2 inhibitors (**d**). **e** TRN1 IRES reporter assay in C4-2 cells; Fluc/Rluc ratio indicates IRES-dependent translation. Data represent Mean ± SD from $n = 4$ biologically independent experiments. **f** Box

plot showing the mRNA level of TRN1 in normal ($n = 52$) and PCa ($n = 497$) specimens (TCGA). $P$ values were calculated by two-tailed unpaired Wilcoxon's test. Box plots as in (**b**). **g** Representative IHC images of TRN1 in benign prostate and PCa. Scale bar: 200 μm. Graph showing the TRN1 protein levels in benign prostate ($n = 114$) and PCa types of untreated ($n = 188$), NHT treated ($n = 34$) and CRPC ($n = 37$). NHT, neoadjuvant hormonal therapy; CRPC, castration-resistant PCa. Box plots as in (**b**). **h** Representative images of PCa TMA slides. Scale bar: 200 μm. **i** Scatter plot showing the correlation between TRN1 and EZH2 protein levels in PCa TMA. Spearman R and P value indicated. Rescue of ADAR1p110 distribution in EZH2-deficient C4-2 cells by ectopic TRN1 expression, shown by western blot (**j**) and GFP fluorescence (**k**, Scale bar: 10 μm). Graphs quantify nuclear/cytoplasmic ADAR1p110 (Mean ± SD, $n = 30$ for siCTRL+EV, $n = 27$ for siEZH2+EV, $n = 28$ for siEZH2 + TRN1). Box plots as in (**b**). Statistical significance was assessed using two-sided student's $t$ test unless otherwise stated. Source data are provided as a Source Data file.

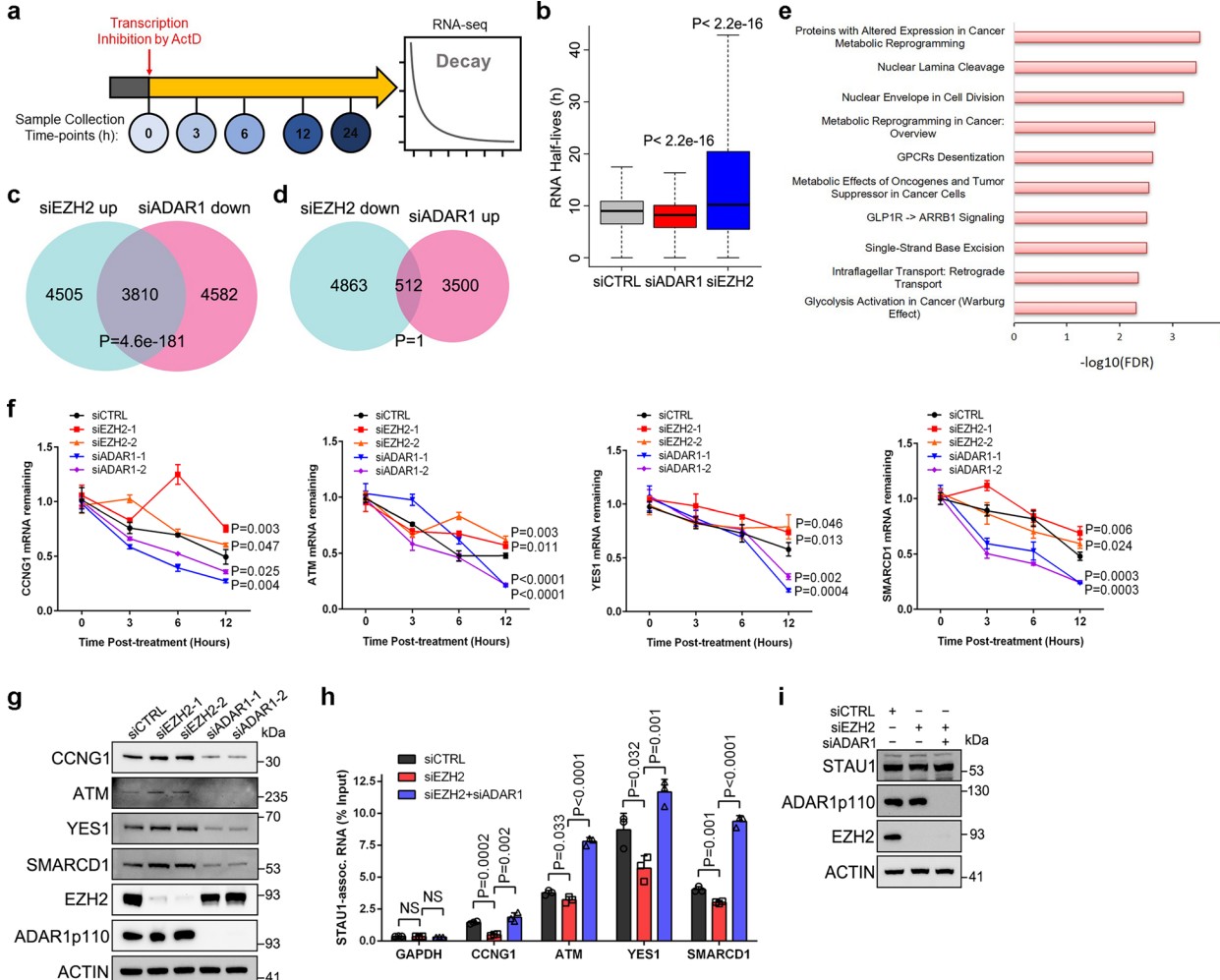

**Fig. 6 | EZH2 deficiency stabilizes a set of oncogenic transcripts through cytoplasmic ADAR1p110. a** The workflow of RNA-seq to measure the global change of RNA half-lives. **b** Box plot showing the RNA half-life in each indicated group based on the RNA-seq data ($n = 2$ for each group). $P$ values were calculated by two-tailed unpaired Wilcoxon's test. Box plots show the median (center line), the interquartile range (box, 25th–75th percentiles), and whiskers indicating the minimum and maximum values. Venn diagrams showing the overlap between gene transcripts with increased stability upon EZH2 deficiency and decreased stability upon ADAR1 deficiency (**c**), or between gene transcripts with decreased stability upon EZH2 deficiency and increased stability upon ADAR1 deficiency (**d**). P values were calculated by one-tailed Fisher's exact test. **e** The selected 3810 gene set (described in **c**) was subjected to Elsevier Pathway Collection enrichment analysis.

**f** The mRNA decay rate of the selected oncogenes was measured by RT-qPCR analysis using samples collected at various time-points after ActD treatment of control, EZH2-deficient or ADAR1-deficient C4-2 cells. Data represent the mean ± SD from $n = 3$ biologically independent experiments. **g** Western blot to detect the protein levels of the selected oncogenes upon EZH2 or ADAR1 knockdown in C4-2 cells. **h, i** RIP-qPCR assay to monitor the binding of selected oncogenic transcripts to STAU1 upon EZH2 knockdown or EZH2/ADAR1 double knockdown in C4-2 cells. GAPDH mRNA containing no dsRNA structure was served as negative control. Data represent the mean ± SD from $n = 3$ biologically independent experiments. The protein level of STAU1 in each group was presented in (**i**). Statistical significance was assessed using two-sided student's t-test unless otherwise stated. Source data are provided as a Source Data file.

3′-UTR dsRNA structures[10], and exhibit oncogenic properties during PCa development[40–43]. RT-qPCR assay was firstly conducted to monitor the remaining mRNA levels at each time-point after ActD treatment. In line with the sequencing results, mRNAs of all tested oncogenes were stabilized upon EZH2 knockdown while destabilized in response to ADAR1 depletion (Fig. 6f). Simultaneously, the steady-state mRNA level of each candidate and the final protein production were concordantly changed following the alterations in mRNA half-lives (Fig. 6g and Supplementary Fig. 6a). To explore whether EZH2 also controls the expression of these oncogenes through conventional transcriptional repression, we labeled and captured nascent RNA from control or EZH2-deficient C4-2 cells, followed by RT-qPCR analysis. As a positive control, the well-known epigenetic EZH2 target CNR1 displayed an accelerated nascent transcription in EZH2-deficient cells. With respect to the tested oncogenes, only SMARCD1 showed a similar pattern with CNR1, whereas the rest candidates were basically unaffected in mRNA synthesis upon EZH2 suppression (Supplementary Fig. 6b). Accordingly, these results confirmed mRNA stabilization as a major contributor for the upregulation of these oncogenes in response to EZH2 depletion. The ADAR1p110-mediated inhibition of mRNA decay relies on its competitive binding to dsRNAs against STAU1[10]. To this end, RNA immunoprecipitation-qPCR (RIP-qPCR) assay was next performed to measure the enrichment of the selected oncogenic transcripts to ADAR1p110 or STAU1 proteins. As presented in Supplementary Fig. 6c, EZH2 knockdown strengthened the binding of ADAR1p110 to all test candidates. In comparison, the enrichment of STAU1 to these candidates were substantially attenuated (Fig. 6h, i). Notably, this reduction was markedly rescued upon additional ADAR1 knockdown, proving that EZH2 deficiency stimulates cytoplasmic ADAR1 to protect these oncogenic transcripts from STAU1 binding and degradation.

## Targeting EZH2 and ADAR1 synergistically inhibit PCa progression

Regardless of the application of EZH2-targeting strategy in a number of clinical trials for advanced PCa treatment[44], our data clearly suggested a limitation of this therapeutics: induces the migration of ADAR1p110 into cell cytoplasm to stabilize a series of prostate oncogenic transcripts. Therefore, it is desirable to evaluate whether targeting ADAR1 for depletion could sensitize cancer cells to EZH2 inhibitors. MS8815, a recently discovered von Hippel-Lindau (VHL)-recruiting EZH2 proteolysis targeting chimera (PROTAC) degrader[45], was employed for EZH2 targeting in our model. By utilizing EPZ6438 as the EZH2 binder while linked with an E3 ligase ligand, MS8815 is capable of selectively targeting EZH2 for ubiquitination and degradation. Treatment of C4-2 cells with MS8815 significantly reduced EZH2 and H3K27me3 levels in a dose-dependent manner (Fig. 7a). Meanwhile, EED and SUZ12 protein levels were decreased synchronously, indicating a disruption of the whole PRC2 complex. In contrast, MS177, another PROTAC-based EZH2 degrader[26], showed a much weaker EZH2 elimination effect in PCa cells (Fig. 7a); while EPZ6438, the parental inhibitor of MS8815, could only erase H3K27me3 marks (Fig. 7b). Moreover, MS8815 exhibited a much lower half-maximal inhibitory concentration ($IC_{50}$) in C4-2 cells as compared with MS177 and EPZ6438 (Fig. 7c), indicating a much more powerful antiproliferative activity. Given that the anti-tumor ability of MS8815 has never been tested in vivo before, we next treated mice bearing C4-2 cells-derived xenograft with a range of MS8815 dosages and monitored the effects. As shown in Supplementary Fig. 7a, b, intraperitoneal (i.p.) injection of MS8815 as low as 25 mg per kg body weight daily was able to trigger tumor repression, and the anti-tumor effect became more significant at the dose of 50 mg per kg body weight per day or above. Meanwhile, all tested doses were well tolerated by mice, as no sudden death or evident weight loss was displayed (Supplementary Fig. 7c). Consistent with our in vitro data, the apparent EZH2 degradation effect was also observed in xenograft tumors (Supplementary Fig. 7d).

To assess the combinational effect of MS8815 with ADAR1 blockade, we first generated a stable C4-2 cell line with inducible expression of ADAR1 shRNA for murine experiments (Supplementary Fig. 7e). As presented in Fig. 7d–f, expression of shADAR1 significantly enhanced the efficacy of MS8815 in mitigating tumor growth. In comparison, EPZ6438 and shADAR1 exhibited no synergistic effect on PCa cell growth, which is compatible with our above conclusion that EZH2 controls ADAR1 shuttling and mRNA stability in a way independent of its enzymatic activity.

To explore the clinical potential of this strategy, we further delivered EZH2/ADAR1 siRNA-containing Nanoparticles (NPs) into LuCaP 35CR PDX-bearing mice for gene silencing (Supplementary Fig. 7f). We found that combinatorial treatment of siEZH2 and siADAR1 significantly improves the clinical outcomes when compared with those utilizing one siRNA only (Fig. 7g, h). In summary, these data suggest that targeting the EZH2-ADAR1 cascade may serve as a more effective strategy than targeting EZH2 alone for PCa treatment.

## Discussion

The evident discordance between the expression level of two enzymatically active ADARs and A-to-I editing frequency has aroused the recent investigations on the non-ADAR factors that could affect the editing process[4]. To date, a number of secondary editing regulators, such as DHX9[46], ILF2/3[36], DAP3[47] and SRSF9[48], have been discovered and functionally characterized. Here, we reported EZH2, a pivotal epigenetic and oncogenic driver, as a distinct ADAR interactor and influential editing modulator in PCa. Despite that EZH2 directly binds to both ADARs in a similar manner, the present study mainly focused on the consequence of EZH2-ADAR1 interaction due to the trace expression of prostatic ADAR2. However, it is expected that the EZH2-ADAR2 interaction may play an indispensable role in various CNS tumor types with abundant expression of both EZH2 and ADAR2[49].

In this study, in-depth transcriptome-wide analysis revealed a bidirectional role of EZH2 in regulation of A-to-I RNA editing. Further mechanistic researches suggested that EZH2 competes with ILF2 for ADAR1 binding, which together contribute to the substrate specificity and selectivity of ADAR1. Interestingly, a previous study demonstrated that ILF2 also affects A-to-I editome in an opposite manner[36], which verified our conclusion from another side. Nevertheless, for multiple reasons, the mechanism underlying EZH2-mediated RNA editing may be far more sophisticated than we currently understood. Firstly, although EZH2 does not modulate ADAR1 expression, it is still possible that EZH2 could indirectly control a number of editing sites by manipulating the expression of other secondary editing regulators canonically or noncanonically; secondly, it is well established that EZH2 inhibitors could strengthen the anti-tumor immune responses by promoting endogenous dsRNA production[50–52]. As a consequence, the enlarged pool of editing substrates may impose an influence on the overall editing pattern; lastly, as mentioned in the latter part of our study, EZH2 depletion leads to the enrichment of cytoplasmic ADAR1p110 to exert its editing-independent function, which is accompanied by a globally diminished editing activity. Accordingly, it is suggested that the landscape of A-to-I editing is remodeled by EZH2 at multiple levels, and our observations on *chr12: 69237519* of *MDM2* transcript may only represent a specific subset of EZH2-mediated editing sites sharing a similar regulatory pathway.

A long-standing question that carries therapeutic potentials is that why quite a high proportion of EZH2-repressed genes (up to 40%, summarized from independent PCa RNA-seq datasets), are actually not targets of H3K27me3 modification[23,53,54]. Herein, we confirmed that, in addition to RNA editing, EZH2 further dictates the editing-independent function of ADAR1p110 as a negative regulator of mRNA decay. Triggered by the translational downregulation of nuclear

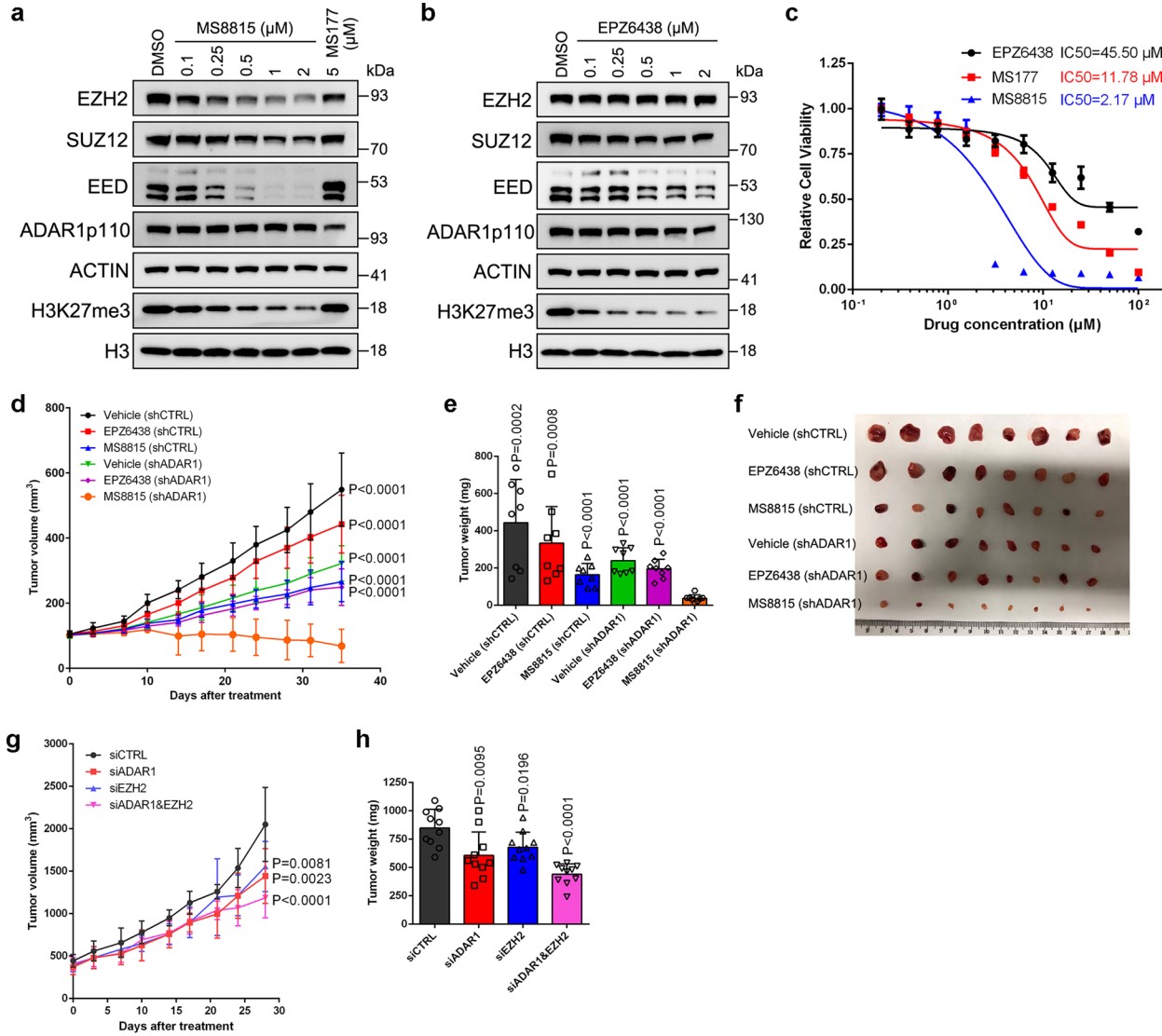

**Fig. 7 | Combinational targeting of EZH2 and ADAR1 reaches a synergistic effect in PCa treatment.** Western blot to detect EZH2, SUZ12, EED, ADAR1 and H3K27me3 levels in C4-2 cells treated with MS8815 (**a**) or EPZ6438 (**b**) at various dosages as indicated. C4-2 cells treated with 5 μM of MS177 was also detected as reference. **c** Growth curve of C4-2 cells treated with MS8815, MS177 or EPZ6438 at different concentrations to measure the IC$_{50}$ value. Data represent the mean ± SD from $n = 6$ biologically independent experiments. **d**–**f** C4-2 cells stably expressing Dox-inducible control or ADAR1 shRNA were injected subcutaneously into SCID mice, followed by treatment using vehicle, MS8815 or EPZ6438. Tumor volume was measured by caliper twice a week and plotted in (**d**). At the end point of measurement, tumors were harvested, weighted (**e**) and pictured (**f**). Data represent Mean ± SD from $n = 8$ mice in each group. **g**, **h** The LuCaP 35CR PDX tumors were implanted subcutaneously into SCID mice, followed by delivery of nanoparticles carrying siRNAs. Tumor volume was measured by caliper twice a week and plotted in (**g**). At the end point of measurement, tumors were harvested and weighted (**h**). Data represent Mean ± SD from $n = 10$ mice in each group. Statistical significance was assessed using two-sided student's $t$ test unless otherwise stated. Source data are provided as a Source Data file.

import receptor TRN1 upon EZH2 deficiency, ADAR1p110 proteins are accumulated in the cytoplasm to protect a subgroup of transcripts with dsRNA structure from degradation. Hence, these target genes, as summarized in Supplementary Data 4, are "derepressed" in the absence of EZH2 not because of the removal of H3K27me3 marks, but indeed due to the prolonged mRNA half-lives. In a word, both silencing of mRNA synthesis and maintenance of mRNA decay manifest the impact of EZH2 on the abundance of cellular transcripts.

The incomplete tumor-suppressive effect of EZH2 enzymatic inhibitors (i.e., GSK126 and EPZ6438) has ushered in a new wave of explorations for pharmacological targeting both the catalytic and non-catalytic functions of EZH2. Fueled by the PROTAC technology, several small-molecule EZH2 degraders, such as MS177[26], E7[55], YM281[56] and most recently, MS8815[45,57], have been developed for cancer therapy. Our results suggested that MS8815 displays much more potent and

effective EZH2-targeting and antiproliferation effects in PCa cells as compared with MS177 and EPZ6438, and we are the first to verify its therapeutic potentials in vivo using the PCa xenograft model. More importantly, our results confirmed that combinatorial targeting of both EZH2 and ADAR1 could achieve a synergistic effect in advanced PCa treatment, presumably due to the complete blockage of both the oncogenic functions of EZH2 and the unfavorable stabilization of a set of oncogenic transcripts upon EZH2 deficiency. Future efforts should be made for the discovery and clinical application of the in vivo-feasible ADAR1 inhibitors.

In summary, our findings indicate a possible mechanism through which EZH2 modulates both the RNA editing-dependent and -independent activities of ADAR1 (Supplementary Fig. 8), although additional studies will be required to fully establish this relationship and its therapeutic implications.

## Methods

### Ethics statement

This study was conducted in accordance with all relevant ethical regulations. All animal experiments were approved by the Institutional Animal Care and Use Committee (IACUC) of Northwestern University and were performed in accordance with institutional and national guidelines. Publicly available datasets were accessed in compliance with their respective data access policies.

### Cell lines, PDX and PDO models

Human PCa cell line C4-2 was a generous gift from Dr. Leland Chung (Cedars-Sinai), while PC-3 was purchased from ATCC. Both PCa cell lines were grown in RPMI 1640 medium supplemented with 10% FBS. Human Prostate Epithelial Cells (PrEC) were purchased from Lonza and cultured using PrEGM prostate epithelial cell growth medium bullet kit (Lonza). HEK293T cells were purchased from ATCC and maintained in DMEM medium supplemented with 10% FBS. All cell lines and primary cells were cultured at 37 °C and 5% $CO_2$ in a humidified atmosphere, and were authenticated and routinely screened for Mycoplasma.

The LuCaP 35CR patient-derived xenograft (PDX) was kindly provided by Eva Corey (University of Washington).

For patient-derived organoid (PDO), PCa PDX MDA-PCa-174 was maintained in immune-deficient, severe combined immunodeficiency (SCID) male mice[58]. Organoids were established using the method described previously[59]. Briefly, tumors were dissected into small pieces and digested with 5 mg/ml collagenase type II in base medium (Advanced DMEM/F12 with 10 mM HEPES and 2 mM Glutamax) at 37 °C for 1 h with gentle shaking. A single tumor cell suspension was obtained after red blood cell lysis. Single cells were then seeded into a 6-well plate in growth factor-reduced Matrigel with complete human prostate organoid culture medium (B27, N-Acetylcysteine 1.25 mM, recombinant human EGF 5 ng/ml, recombinant human Noggin 100 ng/ml, recombinant human R-spondin-1 500 ng/ml, A83-01 500 nM, recombinant FGF-2 5 ng/ml, prostaglandin E2 1 μM, SB202190 10 μM, Nicotinamide 10 mM, DHT 1 nM, Y27632 10 μM in base medium) to form organoids. Knockdown of EZH2, ADAR1, and ADAR2 were achieved by introducing 100 nM of siRNA per well in a 6-well plate, followed by two consecutive knockdowns. RNA samples were harvested 72 h post-second knockdown. RNA was isolated using the Qiagen miRNA isolation kit, and knockdown was confirmed by RT-qPCR before sending samples for sequencing.

### Transfection of siRNA or miRNA mimic

All silencer-select siRNAs used in this study were purchased from Thermo Fisher (siEZH2-1: s4916, siEZH2-2: s4917; siADAR1-1: s1007, siADAR1-2: s1008; siADAR2-1: s1010, siADAR2-2: s1011; siFBL-1: s4820, siFBL-2: s4821; siILF2-1: s7398, siILF2-2: s7399, siTRN1: s7932). In addition, a siRNA targeting the 3′-UTR of endogenous EZH2 (sequence: 5′-UUGCCUUCUCACCAGCUGC-3′) was synthesized by Thermo Fisher and used for the rescue assay. Meanwhile, all mirVana miRNA mimics were purchased from Thermo Fisher (miR-204 mimic: MC11116, miR-211 mimic: MC10168).

Lipofectamine RNAiMAX (Invitrogen) were utilized for siRNA or miRNA mimic transfection according to the procedure of manufacturer. Mediums were renewed after 24 h and cell samples were collected at 3 days post-transfection.

### Treatment of cells with inhibitors

For EZH2 inhibitors, MS8815 and MS177 were discovered and kindly provided by Dr. Jian Jin (Mount Sinai); GSK126 was purchased from BioVision; DZNep and EPZ6438 were purchased from Selleck Chemicals.

### Western blot (WB)

To denature proteins, cell lysates were mixed with 2 × loading buffer (Bio-Rad) and heated at 95 °C for 10 min. Protein samples were subjected to standard SDS-PAGE and semi-dry transferred to PVDF membranes (Roche). After blocking for 45 min in Tris-buffered saline-Tween 20 (TBST) with 5% nonfat milk, membranes were incubated with primary antibodies for 2 h at room temperature. Then, the membranes were washed and incubated with Clean-Blot IP Detection Reagent (Thermo Fisher, 1:500 dilution, for co-IP WB) or goat anti-mouse/rabbit IgG (H + L)-HRP secondary antibody (GenDEPOT, 1:5000 dilution, for normal WB) for 1 h. The signals were developed using western ECL substrate (Bio-Rad) and captured by a Bio-Rad imaging system. The primary antibodies used for WB in this paper were listed in Supplementary Table 1.

### Real-time (RT)-qPCR analysis

To detect mRNA level, the total RNA was isolated using RNeasy Plus Mini Kit (Qiagen), followed by reverse transcription into cDNA using Maxima H Minus First Strand cDNA Synthesis Kit (Thermo Fisher). The cDNA samples were amplified using Universal SYBR Green Supermix (Bio-Rad) in a QuantStudio 6 Flex Real-time PCR System (GE Healthcare) following manufacturer's instructions. The primers used for RT-qPCR analysis were summarized in Supplementary Table 2. The relative RNA level was calculated using the $2^{-\Delta\Delta Ct}$ method with the Ct values normalized to GAPDH.

To measure miRNA expression, miRNeasy Mini Kit (Qiagen) was used to extract small RNA from cells. The obtained small RNA was reverse transcribed into cDNA using TaqMan miRNA Reverse Transcription Kit (Thermo Fisher). TaqMan PCR Assay was then performed following its guideline from Thermo Fisher. The relative miRNA level was calculated using the $2^{-\Delta\Delta Ct}$ method with the Ct values normalized to U6 snRNA. All probes used for TaqMan qPCR analysis were ordered from Thermo Fisher (miR-204: 000508, miR-211: 000514 and U6 snRNA: 001973).

### Co-immunoprecipitation (co-IP) assay

The whole-cell lysate was prepared by lysing cells in the NP-40 lysis buffer (GenDEPOT) containing protease and phosphatase inhibitor cocktail (Thermo Fisher). After sonication and centrifugation to remove the insoluble materials, cell lysates were pre-incubated with Dynabeads protein A/G (Invitrogen) to remove nonspecific binding. Then, antibodies were mixed into lysates with new-added Dynabeads and incubated at 4 °C overnight. The immune complexes were separated by a magnetic rack and washed three times with lysis buffer. To denature proteins, beads were mixed with 2× loading buffer (Bio-Rad) and heated at 95 °C for 10 min. Protein samples were subjected to western blot for further analysis. For the RNase A treatment, the whole-cell lysates were incubated with RNase A (1 μg/ml, Thermo Fisher) at 37 °C for 10 min before immunoprecipitation.

### Proximity ligation assay (PLA)

The Duolink PLA Fluorescence was performed according to the manufacturer's protocol (Sigma). Briefly, control, EZH2-deficient and ADAR1-deficient C4-2 cells grown on coverslips were fixed by 4% paraformaldehyde for 10 min and permeabilized in phosphate-buffered saline (PBS) containing 0.5% Triton X-100 for 10 min. Subsequently, cells were blocked by blocking solution for 1 h at 37 °C and incubated with mouse anti-EZH2 and rabbit anti-ADAR1 antibodies for 2 h. After washing, the slide was further incubated with PLUS and MINUS PLA probes for 1 h at 37 °C, followed by ligation and amplification. The slide was finally mounted using Duolink In Situ Mounting Medium with DAPI (Sigma) and visualized by confocal microscopy.

### GST pull down

To perform GST pull down assay, 1 μg recombinant GST-tagged EZH2 protein was mixed with 1 μg Flag-tagged ADAR1/2 in 1 ml of NP-40 lysis buffer (GenDEPOT) with protease and phosphatase inhibitor cocktail (Thermo Fisher). The mixture was incubated with pre-washed

Glutathione Sepharose 4B (GE Healthcare) for 1 h at room temperature. Then, the protein-attached beads were pelleted by centrifugation and washed for three times. To prepare protein samples, beads were added to 2× loading buffer (Bio-Rad) and heated at 95 °C for 10 min. Protein samples were further analyzed by western blot. The recombinant proteins used were listed in Supplementary Table 3.

## AlphaLISA assay
The AlphaLISA experiment was performed in the dark using white 96-well half area plate (Greiner) according to the User's Guide (PerkinElmer). In brief, protein cross-titration experiment was firstly performed by testing multiple concentration combinations of two proteins in a matrix to detect the optimal analyte association. Next, a third protein or an untagged version of one or the other protein was introduced to the displacement assay to affect the existing association between beads. The alpha signals indicating the binding affinity were measured in a Tecan plate reader using the AlphaLISA detection mode. All the tagged or untagged recombinant proteins used for this assay were summarized in Supplementary Table 3 and the corresponding Donor and Acceptor beads were selected based on the tags of protein pairs.

## RNA sequencing (RNA-seq) and whole genome sequencing (WGS) to identify editing events
Total RNA was isolated from C4-2 cells using RNeasy Plus Mini Kit (Qiagen) and subjected to BGI for Poly(A) selection library preparation and sequencing using the DNBseq platform. Meanwhile, the genomic DNA was extracted from C4-2 cells using Quick-DNA Kit (Zymo Research) and subjected to BGI for human true PCR-free 30x coverage WGS.

For WGS, the raw fastq files were mapped to the human reference genome hg19 with BWA (v0.7.17)[60]. Sorted bam file was then used as the corresponding genome reference of C4-2 for identification of RNA editing events.

For RNA-seq, the raw reads with fastq format were aligned to the human reference genome hg19 from iGenome (https://support. illumina.com/sequencing/sequencing_software /igenome.html) using STAR aligner (v2.7.9a)[61]. The count number of mapped reads was calculated by htseq-count (v0.11.2)[62]. After mapping, the software of REDItoolDnaRna.py from REDItools (v1.3)[63] was applied to detect the RNA editing events using the parameters provided by a published protocol[64]. The C4-2 genome information, database of SNPs (dbSNP) and several stringent filters were utilized to minimize the detection of false RNA editing candidates and obtain high confident editing events. These filtering processes include considering quality score >25, mapping quality >30, base coverage >10× and removal of splicing sites and PCR duplicates. The altered RNA editing events between siADAR1 or siEZH2 and siCTRL were determined using the log likelihood ratio model REDIT-LLR in R package[65]. Only the A-to-I editing events were further recruited for the differential analysis and functional annotations. An editing level cutoff of ≥5% was applied, and only sites with a $P < 0.05$ were considered significant.

Similar analysis was also conducted on Stand Up To Cancer (SU2C) metastatic PCa samples using transcriptome sequencing and matched whole exome sequencing data from the dbGAP dataset (phs000915.v2p2[32]), as well as on PDO samples and C4-2 cells treated with EPZ-6438 and GSK-126 inhibitors. For SU2C data analysis, we divided PCa patient samples into EZH2-high and -low groups using a previous method[66]. The EZH2-knockdown datasets of LNCaP[67], LNCaP-abl[67], HL-60[68], MCF-7[69], and SUM159[70] cells were downloaded from GEO database. The REDItools de novo analysis[63] was conducted to detect and quantify RNA editing levels in these samples.

## RNA editing analysis using TCGA samples
RNA-seq alignments stored as BAM files from 552 prostate adenocarcinoma (PRAD) patients were downloaded through gdc-client v1.6.1. To associate the biospecimen data within TCGA barcodes, the GenomicDataCommons package v1.22.1[71] was further used to map from the universal unique identifier (UUID). For each biospecimen, variant calling was performed using SAMtools v1.5 mpileup[72] and bcftools v 1.16 call[73]. Filtering was performed by discarding SNVs or small indels in which the variant calling score at the QUAL field is lower than 20. The editing ratio of *chr12:69237512* was calculated using the VCF file of the TCGA primary prostate dataset and the normal prostate dataset from RNA editing database[74].

## RNA editing site-specific quantitative RT-PCR (RESSq-PCR) assay
The RESSq-PCR approach is a sensitive RNA editing fingerprint assay to detect RNA recoding[33]. To conduct this, RNA editing site-specific primers that were compatible with SYBR green RT-qPCR assay were firstly designed based on the principle of this strategy and were presented in Supplementary Table 2. Then, the RT-qPCR analyses were carried out using the cDNA samples from each group. The relative RNA editing level (Edited/WT ratio) was calculated as $2^{(CtWT-CtEdited)}$.

## Sanger sequencing
The MDM2 cDNA fragment containing the interested editing site was PCR-amplified using amfisure PCR Master Mix (GenDEPOT) and purified using QIAquick PCR Purification Kit (Qiagen). The obtained PCR products were shipped to GENEWIZ for Sanger sequencing. Sequencing chromatograms were visualized using SnapGene Viewer and the A-to-G editing frequency was calculated as %G/(A + G) based on the raw sequencing data. The primer sequences were presented in Supplementary Table 2.

## MDM2 3'-UTR reporter assay
Both control (CmiT000001-MT06) and wildtype MDM2 3'-UTR (HmiT102206-MT06) reporter clones were purchased from GeneCopoeia, while the edited MDM2 mutant was constructed through A-to-G site-directed mutagenesis. HEK293T cells overexpressing control, miR-204 or miR-211 mimics were seeded into 24-well plate and transfected with control, wildtype or edited 3'-UTR reporter plasmids. Dual-luciferase assays were performed at 24 h post-transfection of reporters using Luc-Pair Duo-Luciferase HS Assay Kit (GeneCopoeia) according to the manufacturer's instructions. A Tecan plate reader was used to measure the bioluminescence intensity.

## Label-free quantitative co-IP/MS
To quantitative comparison of proteins, duplicates of control and EZH2-deficient C4-2 cells were subjected to nuclear isolation using NE-PER nuclear extraction kit (Thermo Fisher), followed by co-IP assay to pull down ADAR1 along with its interacting proteins using the method as mentioned above. IP eluate of each sample was then run into the stacking gel and submitted to Northwestern Proteomics Core for MS analysis by following a previously established protocol[75].

## Cell fractionation assays
For nuclear and cytoplasmic extraction, NE-PER nuclear extraction kit (Thermo Fisher) was utilized to separate the nuclear and cytoplasmic fractions of C4-2 cells, followed by western blot analysis.

For size-exclusion chromatography, nuclear extracts from control and EZH2-deficient C4-2 cells were prepared using NE-PER nuclear extraction kit (Thermo Fisher), and 5 mg of obtained nuclear protein was concentrated into 200 μL using Ultra-0.5 Centrifugal Filter (Amicon). The concentrated sample was then loaded into Pharmacia SMART system for size-exclusion. A total of 48 fractions with a volume of 50 μL for each fraction were finally obtained and subjected to western blot assay.

## Enhanced crosslinking and immunoprecipitation-sequencing (eCLIP-seq)
The eCLIP-seq experiment was performed in duplicates following the method described previously[76]. For each sample, a total of 50 million

C4-2 cells were crosslinked, lysed and partially digested by RNase I (Thermo Fisher). The resulting lysates were incubated with the antibody of interested RBP and Protein A/G magnetic beads (Pierce) at 4 °C overnight, followed by washing, end repair and 3′ adaptor ligation. The protein-RNA complexes were then eluted from the beads, resolved by denaturing gel electrophoresis and transferred to a nitrocellulose membrane. Subsequently, the region of interest was cut from membrane and subjected to RNA extraction, cDNA synthesis and library preparation. The cDNA library was further amplified using Q5 PCR Master Mix (NEB) and sent to BGI for sequencing using the DNBseq platform.

For analysis, both STAR (v2.7.9a)[61] with default parameters and UCSC known genes were firstly employed to map the genome hg19, followed by peak calling using CLIP-seq peak caller Piranha with default parameters and bin size=20[77]. To define the RBP binding peaks in each sample, a P-value cutoff of 0.01 was set up. The homer (v4.8.3) annotatePeaks.pl script[78] was then applied to annotate the peaks from the eCLIP-seq data. A down sampling step was performed using the samtools view function to match the sample with the smallest read count. The software bamCoverage from deeptools (v3.5.5)[79] was used to generate a coverage track bigWig as output, the bigwigCompare was used to generate the input subtracted bigwig signal.

### Immunofluorescence

C4-2 cells were cultured on coverslips for 24 h before immunofluorescence staining. After fixation in pre-cooled methanol for 15 min, cells were permeabilized in PBS containing 0.5% Triton X-100 for 10 min at room temperature. Slides were rinsed thrice with PBS and blocked with 5% BSA in PBS for 30 min. Then, the slides were co-incubated with rabbit anti-EZH2 antibody and mouse anti-GFP antibody at 4 °C overnight. After rinsed thrice with PBS, the slides were exposed to Alexa Fluor 488 conjugated goat anti-mouse and Alexa Fluor 555 conjugated goat anti-rabbit antibodies (1:200, Invitrogen) for an additional hour at 37 °C, followed by three washes with PBS. Cell nuclei were stained in the dark with DAPI (Invitrogen) and mounted using ProLong Diamond Antifade Mountant (Invitrogen). Immunostained cells were viewed and photographed using Nikon A1R confocal microscope. The primary antibody information could be found in Supplementary Table 1.

### Dual-luciferase assay to detect IRES activity of TRN1

The bi-cistronic luciferase vector containing IRES element of TRN1 was constructed as described previously[27], while the proposed TRN1 IRES sequence was determined based on a previous report[39]. The dual-luciferase assay was performed at 24 h post-transfection using the Dual-Glo luciferase reagent (Promega) according to the manufacturer's instructions, followed by signal capture through a Tecan plate reader.

### Tissue microarrays (TMAs) and immunohistochemistry (IHC) staining

Prostate tumor biopsies retrieved from Vancouver Prostate Center tissue bank were recruited to construct TMAs as published previously[80]. This protocol was approved by the office of research ethics in the University of British Columbia. IHC was performed using Ventana Discovery XT autostainer (Ventana) with the indicated antibodies by following its manual. All stained slides were recorded by a Leica SCN400 scanner. Digital images were evaluated and scored by a pathologist, Dr. Ladan Fazli. The histology score (H-score) of each stained protein was calculated by the Aperio ImageScope software (Leica Biosystems) based on both intensity and percentage of the IHC signals. All primary antibodies used for IHC were listed in Supplementary Table 1.

### mRNA stability assay using actinomycin D (ActD)

To analyze the impact of EZH2 or ADAR1 suppression on mRNA decay rate, control, EZH2- or ADAR1-deficient C4-2 cells were incubated with 5 µg/ml ActD (Sigma). Cell samples were collected at 0, 3, 6, 12, and 24 h post-treatment, followed by total RNA isolation using RNeasy Plus Mini Kit (Qiagen). The obtained RNA was either subject to RNA-seq analysis ($n = 2$) in BGI or RT-qPCR analysis ($n = 3$) after reverse transcription.

The RNA half-lives in each sample were calculated using a previously described method[81]. In brief, to minimize the gene expression noise caused by genes that showed higher expression levels after the ActD treatment, we scaled the FPKM values based on the expression levels of the top 10 upregulated genes. The RNA degradation rate $k_{decay}$ was calculated using the average ratio of the RPKM value at 0 h post-treatment and the RPKM values at the other time-points. The pseudo-RNA half-life $t_{1/2}$ was then determined by calculating $\ln(2)/k_{decay}$.

### Isolation and detection of nascent RNA

The newly minted RNAs were enriched using Click-iT Nascent RNA Capture Kit (Thermo Fisher) by following manufacturer's manual. In general, the nascent RNAs were firstly labeled by incubation of live cells with 5-ethynyl uridine (EU) for 6 h. After incubation, the total RNA containing EU-labeled nascent RNA was extracted using RNeasy Plus Mini Kit (Qiagen) and used in a copper catalyzed click reaction with an azide-modified biotin. The captured nascent RNA transcripts on streptavidin magnetic beads were reverse transcribed using the SuperScript VILO cDNA synthesis kit (Thermo Fisher). The obtained cDNAs were then subjected to RT-qPCR analysis with primers summarized in Supplementary Table 2. GAPDH was employed as an internal control.

### RNA immunoprecipitation (RIP)-qPCR analysis

The RIP assay was performed in triplicates using the EZ-Magna RIP kit (Millipore) using the procedure provided by the manufacturer. Control or treated C4-2 cells were lysed using RIP lysis buffer and incubated with antibody-magnetic beads mixture at 4 °C overnight to enrich targeted proteins along with the associated mRNAs. These interacting mRNAs were collected by Proteinase K digestion, followed by purification and reverse transcription into cDNA. Then, RT-qPCR assay was performed to measure the %Input of RBP-binding mRNAs in each group using the primers presented in Supplementary Table 2.

### Cell viability assays

To measure $IC_{50}$ of EZH2 inhibitors, C4-2 cells were seeded into 96-well plates at a density of around 1,000 cells per well. After 24 h, cells were treated with DMSO or indicated serial dilutions of each drug for 3 days. Cell viability was assessed using CellTiter-Glo Luminescent Cell Viability Assay Kit (Promega) and was read on a Tecan plate reader. The cell proliferation curve was drawn and fit by the bioluminescence to drug concentration and the $IC_{50}$ was calculated with non-linear fitting.

### Generation of C4-2 cells stably expressing inducible ADAR1 shRNA

The ADAR1 shRNA oligos were synthesized in IDT with the sequences shown in Supplementary Table 2. The oligos were annealed and ligated into pLKO-Tet-On vector (Novartis). The obtained pLKO-Tet-On shADAR1 construct and the helper plasmids were then co-transfected into HEK293T cells to pack lentiviruses, followed by infection into C4-2 cells. The stable Dox-inducible shADAR1-expressing C4-2 cell line, along with the negative control cell line, were finally obtained by selection with puromycin (Sigma) and verified by western blot analysis.

### Murine PCa xenograft model

Five-week-old male SCID mice were purchased from Charles River. Animal care and use conditions were followed in accordance with institutional and National Institutes of Health protocols and guidelines,

and all studies were approved by Northwestern University Animal Care and Use Committee. The maximal permitted tumor burden was 1000 mm³, as defined by the approved protocol. Tumor volumes were monitored regularly, and animals were euthanized before reaching this limit. The maximal tumor size/burden permitted by the IACUC was not exceeded in any experiment.

MS8815 was dissolved in 5% (vol/vol) NMP, 5% (vol/vol) Solutol HS-15 and 90% (vol/vol) normal saline; while EPZ6438 was dissolved in ddH$_2$O containing 0.5% NaCMC and 0.1% Tween-80. Both drugs were administered twice a day.

For drug optimization assay, each mouse was injected subcutaneously in the left flank with $2 \times 10^6$ C4-2 cells suspended in 100 μL of PBS with 50% Matrigel (BD Biosciences). After tumor volumes reached above 50 mm³, mice were randomized to 5 groups of 4 mice each: (1) vehicle control; (2) 25 mg/kg/day MS8815; (3) 50 mg/kg/day MS8815; (4) 100 mg/kg/day MS8815; (5) 200 mg/kg/day MS8815. Drugs were intraperitoneally (IP) injected 5×/week for 4 weeks. Tumor size was recorded twice a week over the course of the studies using calipers and tumor volume was calculated as follows: MIN(a)$^2 \times$ MAX(b) $\times$ 0.5. Animal body weights were measured at the beginning and the end of the treatment. After 28 days of treatment, mice were euthanized, tumors were excised and weighed. The effect of MS8815 treatment in degradation of EZH2 was further examined by western blot.

For combination assay, each mouse was injected subcutaneously in the left flank with $2 \times 10^6$ Dox-inducible shCTRL or shADAR1 C4-2 cells suspended in 100 μL of PBS with 50% Matrigel (BD Biosciences). When tumor volumes reached 100 mm³, doxycycline (Sigma) was added to the drinking water (final concentration, 100 μg/ml) and the two sets of mice were further assigned into 6 groups of 8 mice each: (1) vehicle-shCTRL; (2) EPZ6438-shCTRL (150 mg/kg/day); (3) MS8815-shCTRL (50 mg/kg/day); (4) vehicle-shADAR1; (2) EPZ6438-shADAR1 (150 mg/kg/day); (3) MS8815-shADAR1 (50 mg/kg/day). Drugs were delivered either by oral gavage (EPZ6438) or IP injection (MS8815) 5×/week for 5 weeks. Tumor size was routinely recorded as described above. After 35 days of treatment, mice were euthanized, tumors were excised and weighed.

For the siRNA-loaded NP animal assay, the bulk EZH2 (sequence: 5′- GCUGACCAUUGGGACAGUAUU-3′) and ADAR1 (sequence: 5′- CAGUGUUCCUGAAACCGCUUU-3′) siRNAs were synthesized in Dharmacon along with the cy5-Luciferase control siRNAs. The lipid-polymer hybrid siRNA NPs were formulated by a self-assembly nanoprecipitation method with modifications[82]. Briefly, 5 mg PLGA and 0.5 mg G0-C14 were dissolved in 1 mL of organic solvent (e.g., DMF) respectively. 50 μL of siRNA (4 nM) aqueous solution were mixed with the G0-C14 organic solution by pipetting to form a siRNA/G0-C14 nanocomplex. Next, the organic solution with nanocomplexes and polymers was added dropwise to 20 mL of aqueous solution containing 2 mg of lipid-PEG (e.g., DSG-PEG). 0.2 mL of PBS was subsequently added to stabilize the formed NPs. The suspension was then stirred at room temperature for 30 min. The hybrid nanoparticles were finally washed in Amicon tubes (MWCO 100 kDa; Millipore), the remaining organic solvent and free compounds were removed with ice-cold water, and concentrated in 1 mL of PBS solution.

LuCaP 35CR PDX bits were implanted subcutaneously into 6–8 weeks old pre-castrated male athymic SCID mice. Mice were monitored for up to 12 weeks post implantation for initial growth. When the tumors were approximately 200 mm³ in size, mice were randomized into treatment groups. Tumor size was measured three times weekly using the formula mentioned above. Intratumoral delivery into the subcutaneous flank tumors were administered twice a week for 3 weeks in 50 μL volume with 28 Gauge needle syringe. Sonication is performed to achieve multiple purposes including deagglomeration of nanoparticles every time before treatment. Mice were euthanized when tumor volume exceeded 1000 mm³ or at the end of a predefined experimental endpoint.

## Visualization of protein domain structures
The domain structure of protein was illustrated by Domain Graph (DOG) 2.0 software[83].

## Statistics and reproducibility
Statistical analysis was performed using GraphPad Prism (version 6.0) or R and presented as means ± SD. Unless otherwise specified, the $P$ values were obtained using two-tailed Student's t-tests for comparison of two datasets or by analysis and variance (ANOVA) where appropriate. Statistical data were considered significant if $P < 0.05$. The results were reproducible and conducted with established internal controls. When feasible, experiments were repeated three or more times and yielded similar results. We have indicated the n values used for each analysis in the figure captions.

## Reporting summary
Further information on research design is available in the Nature Portfolio Reporting Summary linked to this article.

## Data availability
The data supporting the findings of this study are available from the corresponding authors upon request. The next-generation sequencing data have been deposited in the Gene Expression Omnibus (GEO) under accession code of GSE225951. Source data for the figures and Supplementary Figs. are provided as a Source Data file. Source data are provided with this paper.

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

## Acknowledgements

We thank Center for Advanced Microscopy/Nikon Imaging Center of Northwestern University for assistance with confocal microscopy. We thank Vancouver Prostate Center for assistance with IHC assay. Proteomics services were performed by the Northwestern Proteomics Core Facility, generously supported by NCI CCSG P30 CA060553 awarded to the Robert H Lurie Comprehensive Cancer Center, instrumentation award (S10OD025194) from NIH Office of Director, and the National Resource for Translational and Developmental Proteomics supported by P41 GM108569. This work was supported in part by a startup funding provided by the Northwestern University. Part of effort for Q.C. was supported by Northwestern University, American Cancer Society (RSG-15-192-01), U.S. Department of Defense (W81XWH-17-1-0357, W81XWH-19-1-0563 and W81XWH-20-1-0504), NIH/NCI (R01CA256741, R01CA285684, R01CA300246, R01CA278832 and Prostate SPORE P50CA180995 Development Research Program) and Polsky Urologic Cancer Institute of the Robert H. Lurie Comprehensive Cancer Center of Northwestern University at Northwestern Memorial Hospital. K.C. is supported by NIH R01GM138407, R01GM125632, R01HL148338, and R01HL133254. Y.Y. was supported by U.S. Department of Defense grant HT9425-23-1-0661, NIH P50CA180995 SPORE in Prostate Cancer Career Enhancement Award, and The Elsa U. Pardee Foundation Research Grant. X.D. was supported by the Canadian Institute of Health Research (CIHR) project grants (PJT156150, PTJ178063) and DoD prostate cancer ideal award (PC190327). Jinjun.S. was supported by NIH grant R01CA200900. L.W. was supported by NIH grant R35GM146979, and the Research Scholar Grant (RSG-22-039-01-DMC) from the American Cancer Society. H.N. was supported by NIH grant GM124765 and American Cancer Society Research Scholar award RSG-21-013-01-DMC. H.H. was supported by NIH grant R01GM130759. P.M. was supported by NIH grants R37CA258730, R01CA288820, R01CA292949 and U.S. Department of Defense W81XWH21-1-0520; X.L. was supported by U.S. Department of Defense W81XWH2110418.

## Author contributions

Y.Y. and Q.C. conceived and designed the research with the help of W.Z., H.A.H., and E.M.S.; Y.Y. performed a majority of the experiments with assistance from R.W., Qi.L., C.Y., Qiaqia.L, S.W., X.L., and P.M.; X.Y. synthesized the EZH2 degraders of MS8815 and MS177 under supervision of J.J.; Y.Q. performed the PCa PDO-related experiments under supervision of A.M.C.; A.S. conducted the size-exclusion chromatography under supervision of L.W.; L.F. and X.D. performed the IHC assay; Y.Z. prepared the nanoparticles under supervision of Jinjun.S.; Jiangchuan.S. constructed several EZH2 mutants under supervision of H.N.; Y.L. and X.W. conceived, designed, and performed bioinformatics analysis under supervision of K.C.; Y.Y. wrote the paper; All authors discussed the results and commented on the manuscript.

## Competing interests

J.J. is a cofounder and equity shareholder in Cullgen, Inc., a scientific cofounder and scientific advisory board member of Onsero Therapeutics, Inc., and a consultant for Cullgen, Inc., EpiCypher, Inc., Accent Therapeutics, Inc, and Tavotek Biotherapeutics, Inc. The Jin laboratory received research funds from Celgene Corporation, Levo Therapeutics, Inc., Cullgen, Inc. and Cullinan Oncology, Inc. A.M.C. is a co-founder and serves on the scientific advisory boards of LynxDx, Oncopia and Esanik. A.M.C. serves on the scientific advisory board of Tempus and Ascentage. The other authors declare no competing interests.

## Additional information

[1]Department of Urology, Feinberg School of Medicine, Northwestern University, Chicago, IL, USA. [2]Robert H. Lurie Comprehensive Cancer Center, Northwestern University Feinberg School of Medicine, Chicago, IL, USA. [3]Basic and Translational Research Division, Department of Cardiology, Boston Children's Hospital, Boston, MA, USA. [4]Department of Pediatrics, Harvard Medical School, Boston, MA, USA. [5]Prostate Cancer Program, Dana-Farber Harvard Cancer Center, 450, Brookline, MA, USA. [6]Mount Sinai Center for Therapeutics Discovery, Departments of Pharmacological Sciences and Oncological Sciences, Tisch Cancer Institute, Icahn School of Medicine at Mount Sinai, New York, NY, USA. [7]Center for Nanomedicine and Department of Anesthesiology, Perioperative and Pain Medicine, Brigham and Women's Hospital, Harvard Medical School, Boston, MA, USA. [8]Michigan Center for Translational Pathology, University of Michigan, Ann Arbor, MI, USA. [9]Department of Pathology, University of Michigan, Ann Arbor, MI, USA. [10]Rogel Cancer Center, University of Michigan, Ann Arbor, MI, USA. [11]Department of Biochemistry and Molecular Genetics, Feinberg School of Medicine, Northwestern University, Chicago, IL, USA. [12]Vancouver Prostate Centre, Vancouver General Hospital, Vancouver, BC, Canada. [13]Department of Urologic Sciences, University of British Columbia, Vancouver, BC, Canada. [14]Department of Molecular and Cellular Biochemistry, Indiana University, Bloomington, IN, USA. [15]Department of Urology, School of Medicine, Yale University, New Haven, CT, USA. [16]Department of Biology, Indiana University, Bloomington, IN, USA. [17]Department of Urology, University of Michigan, Ann Arbor, MI, USA. [18]Howard Hughes Medical Institute, University of Michigan, Ann Arbor, MI, USA. [19]Center for Inflammation and Epigenetics, Houston Methodist Research Institute, Houston, TX, USA. [20]These authors contributed equally: Yang Yi, Yanqiang Li.
✉e-mail: kaifu.chen@childrens.harvard.edu; qi.cao@northwestern.edu

