## [Transparent Peer Review file · Nature Communications]

A dual role of EZH2 in regulating A-to-I RNA editing and mRNA stability through ADAR

Corresponding Author: Dr Qi Cao

Version 0:

Reviewer comments:

Reviewer #4

(Remarks to the Author)

Overall I find this to be a difficult to follow study and mechanism and one that has not revealed itself in physiological conditions based on many mouse models and cell lines models of ADAR1 and EZH2 biology.

Specific comments:

- all westerns should be quantitated and statistically assessed.
- no analysis of the transcriptional changes that occur with knock-down of Ezh2 compared to ADAR1 - are the same pathways being regulated or is there little overlap? They are likely to be significantly different between siADAR1 and siADAR2 so the comparison to siEzh2 would be enlightening as to how much of the transcriptional signature of siADAR1 is shared.
- what type of RNA-seq was done - it says BGI undertook it but this doesn't explain whether it was ribosome depleted or polyA libraries. Based on the distribution of editing sites I assume it is most likely polyA as the relative proportions of repeat editing are generally much higher than shown in Fig 2c.
- The authors must specify which ADAR1 isoform they are talking about where. This is a profound deficiency in the current text.
- related to ADAR1 isoforms - it is not clear that any of the reagents used can specifically delineate between ADAR1p110 and ADAR1p150 - this applies to the siRNAs and the antibody used for localisation of the suggested "cytoplasmic" ADAR1. If the reagent can't discern between the ADAR1 isoforms then it is hard to substantiate that it's p110 (as implied) that is in the cytoplasm.
- 8-Azaadenosine is not an ADAR1 inhibitor - <https://pubmed.ncbi.nlm.nih.gov/35586115/>
- Data don't support the conclusion in lines 559-561 - this work presents very limited and a potentially subtle effect of Ezh2 on ADAR1 activity.
- it is incorrect to describe the editing of a specific single adenosine as characterised for the editing chr12: 69237519 as hyperediting (Hyperediting is most commonly used in the field to refer clusters of editing in the same transcript not increases in editing at a specific site - see <https://www.nature.com/articles/ncomms5726>).

Comment on the authors' response to the previous Reviewers

- the adjusted Venn diagrams in the response to reviewers comments should be presented in the main figure.
- I find the text presented in response to reviewer 1 point 3 unconvincing and the eCLIP data as presented show broad enough binding patterns that overlaps may be circumstantial;
- the response to reviewer 1 point 4 should be conducted with the same window and preferably a smaller one as eCLIP should give residue level information about the RNA binding footprint of each when done correctly (200nt for ADAR1 and EZH2 or 1000nt for both - it is not clear why a different window of binding would be appropriate or justified).
- The model proposed involving all three dsRBD of ADAR1 should be supported with AlphaFold predictions if possible - I find it hard to see how ADAR1 could commit all three dsRBD to interacting with EZH2 and there be both increased and decreased editing. The alternative model could be that the interaction occurs separately from editing by ADARs.

-There are multiple examples of unexpected grammar which also impair readability (e.g. line 529 - " Accordingly, it is esteemed that the landscape of ..." - esteemed is a strange choice)

Version 1:

Reviewer comments:

Reviewer #4

(Remarks to the Author)

The authors have made efforts to amend the manuscript in response to the comments raised during review. It remains a very dense and complex manuscript with the chosen figure presentation not helping the reader access the important findings given how dense each figure panel is.

Based my initial reviews and considering both the response to reviewers from the authors and the changes to the text itself I would recommend removal of the 8-Aza data. It has been demonstrated not to be an ADAR inhibitor so why the authors have decided to persist as if it is is not clear to me at all. The moderated text is the the manuscript wants the reader to conclude that 8-Aza is an in cell ADAR editing regulator when this is clearly not established or supported by the data available from cell line treatments with this agent (which notably are not cited by the present authors). The current text is disingenuous when the authors acknowledge in the rebuttal that 8-Aza is not an inhibitor and then make a superficial change to the text.

The authors should clearly state the level of editing change considered significant for the analysis (ie what cut off was used to call a site edited and then what change (5%, 10% etc) was considered significant and how many of the replicates were required to support the changes).

Why has Figure 4a have statistical analysis when it says n=2?

The meaning of "edited/Wt ratio" for the analysis of the editing in Figure 2g-2h needs to be more clearly defined. A change from 1% editing to 2% editing is a 2 fold difference but unlikely to be meaningful – the way the data has been chosen to be displayed removes the important information about the absolute level of editing (as shown in Fig 3a) of a given site from the analysis.

Line 361 – no primary reference to support editing-independent p110 functions in the cytoplasm? Cited reference is a general review

Line 206-208 – no evidence genetically for compensation in vivo – this claim is not required.

Version 2:

Reviewer comments:

Reviewer #4

(Remarks to the Author)

I have no additional comments on the manuscript.

Point-by-point response to Reviewer's comments

Dear Reviewer,

We would like to thank you for taking the time to provide critical and insightful comments on our manuscript. Your suggestions are very helpful for us to improve our research. We have carefully considered the comments and done our best to address every one of them. We hope you will appreciate our strong new supporting results and efforts in this revised version of manuscript.

At the following, all points mentioned by the reviewer will be discussed.

Reviewer #4:

Overall, I find this to be a difficult to follow study and mechanism and one that has not revealed itself in physiological conditions based on many mouse models and cell lines models of ADAR1 and EZH2 biology.

Response: We appreciate the reviewer's thoughtful comment and the opportunity to clarify our study. In this work, we have evaluated the proposed mechanism across multiple *in vitro* and *in vivo* systems, including cancer cell lines, primary normal cells, patient-derived xenografts (PDXs), and patient-derived organoids (PDOs). We recognize that the mechanistic interaction between ADAR1 and EZH2 is complex and cannot be fully delineated within a single study. Accordingly, we have carefully revised the manuscript to temper our claims and to focus on the findings that are directly supported by our data.

Specific comments:

- all westerns should be quantitated and statistically assessed.

Response: Thanks for pointing out this issue. In the revised manuscript, we have quantitated and statistically assessed the key western blot data in **Fig. 3i**, **Fig. 4b**, **Fig. 5a**, **Fig. 5j**, **Fig. S4f** and **Fig. S5a**. We have updated all related figures to show these quantification results.

- no analysis of the transcriptional changes that occur with knock-down of Ezh2 compared to ADAR1 - are the same pathways being regulated or is there little overlap? They are likely to be significantly different between siADAR1 and siADAR2 so the comparison to siEzh2 would be enlightening as to how much of the transcriptional signature of siADAR1 is shared.

Response: Following the reviewer's instructive suggestion, we performed a comparative analysis of differentially expressed genes (DEGs) upon EZH2, ADAR1, or ADAR2 knockdown in C4-2 cells. As shown in **Response Letter Figure 1**, both upregulated and downregulated genes in these groups exhibit significant overlap and are enriched in multiple biological pathways that are either shared or unique. This observation is expected, as RNA editing can modulate mRNA targets through diverse mechanisms that do not always correspond directly to transcriptional changes. Meanwhile, EZH2 is a well-established master transcriptional regulator in cancer cells, and its depletion can cause extensive transcriptomic reprogramming. Therefore, it is

challenging to meaningfully assess the shared transcriptional signatures between siADAR1/2 and siEZH2 conditions.

- what type of RNA-seq was done - it says BGI undertook it but this doesn't explain whether it was ribosome depleted or polyA libraries. Based on the distribution of editing sites I assume it is most likely polyA as the relative proportions of repeat editing are generally much higher than shown in Fig 2c.

Response: We appreciate the reviewer's question. Yes, we performed Poly(A) selection-based RNA-seq. We have revised the **Methods** section to clarify this point.

- The authors must specify which ADAR1 isoform they are talking about where. This is a profound deficiency in the current text.

Response: Thanks for pointing out this potential weakness. In the revised manuscript, we have carefully specified the ADAR1 isoform being referred to throughout the text and figures. In this study, we mainly studied the ADARp110 isoform since this is the major isoform expressed in prostate cancer cell lines and tissues (please see the details below).

- related to ADAR1 isoforms - it is not clear that any of the reagents used can specifically delineate between ADAR1p110 and ADAR1p150 - this applies to the siRNAs and the antibody used for localization of the suggested "cytoplasmic" ADAR1. If the reagent can't discern between the ADAR1 isoforms then it is hard to substantiate that its p110 (as implied) that is in the cytoplasm.

Response: We thank the reviewer for this thoughtful comment. We agree that most commercially available siRNAs and antibodies, including those used in our study, recognize both ADAR1 isoforms. However, several lines of evidence from our study indicate that the cytoplasmic ADAR1 species we focused on corresponds to the p110 isoform:

1- In prostate cancer (PCa) cells, we consistently observed a single ADAR1 band corresponding to the p110 isoform (as shown in all related uncropped western blot images). The p150 isoform was only detectable in normal PrEC cells (**Fig. S2j**).

2- Our western blot analyses (**Fig. 5a and Fig. 5j**) demonstrated that upon EZH2 depletion, only the ADAR1p110 isoform, not p150, was redistributed from the nucleus to the cytoplasm.

3- Overexpression of GFP-tagged ADAR1p110 in PCa cells further confirmed that this isoform was translocated to the cytoplasm following EZH2 knockdown (**Fig. 5b**), and can be rescued by re-expression of TRN1 (**Fig. 5k**).

Together, these results support our conclusion that the cytoplasmic ADAR1 observed in this study corresponds to the p110 isoform.

-8-Azaadenosine is not an ADAR1 inhibitor-
<https://pubmed.ncbi.nlm.nih.gov/35586115/>

Response: We thank the reviewer for raising this important point. We agree with the conclusion of the cited study that 8-aza is not a specific ADAR1 inhibitor. Therefore, we only used this compound in certain cell line-based assays and relied on ADAR1-specific shRNA/siRNA approaches to validate our findings *in vivo*. Indeed, to date, no selective ADAR1 inhibitor is available. Recently, one group reported the development of a small-molecule ADAR1 inhibitor, ZYS-1[1]. However, multiple follow-up studies have shown that ZYS-1 treatment neither reduces cellular A-to-I editing nor downregulates ADAR1 expression[2, 3]. Consistent with these reports, when we tested ZYS-1 in our A-to-I editing reporter C4-2 cell line, we found that 8-aza, but not ZYS-1, significantly decreased the editing level (**Response Letter Figure 2A and 2B**). Based on these results, we believe that 8-aza remains the most suitable compound for *in vitro* studies at this stage. To avoid misunderstanding, we have revised our

description of 8-aza from “ADAR1 inhibitor” to “RNA editing repressor.”

- Data don't support the conclusion in lines 559-561 - this work presents very limited and a potentially subtle effect of Ezh2 on ADAR1 activity.

Response: We have tempered the related claims in the manuscript into “In summary, our findings indicate a possible mechanism through which EZH2 modulates both the RNA editing-dependent and -independent activities of ADAR1 (**Supplementary Fig. 8**), although additional studies will be required to fully establish this relationship and its therapeutic implications” to address this concern.

- it is incorrect to describe the editing of a specific single adenosine as characterised for the editing chr12: 69237519 as hyperediting (Hyperediting is most commonly used in the field to refer clusters of editing in the same transcript not increases in editing at a specific site - see <https://www.nature.com/articles/ncomms5726>).

Response: Thanks for pointing out this error. In the revised manuscript, we have changed “hyperediting” into “increased editing level” or “over-editing” in all related text.

Comment on the authors' response to the previous Reviewers

-the adjusted Venn diagrams in the response to reviewer's comments should be presented in the main figure.

Response: Thanks for the suggestion. Due to the size limit of **Fig. 2**, we have moved these adjusted Venn diagrams to **Supplementary Fig. 2g-2i** and revised the text accordingly.

-I find the text presented in response to reviewer 1 point 3 unconvincing and the eCLIP data as presented show broad enough binding patterns that overlaps may be circumstantial;

Response: Thank you for this important question. Inspired by your subsequent comment regarding the overlap between eCLIP peaks and RNA editing sites, we analyzed the relationship between ADAR1/EZH2/ILF2 eCLIP peaks and EZH2-mediated editing sites. We found that, as compared with ILF2, EZH2-binding regions preferentially overlap with the under-edited sites. In contrast, the ILF2-binding regions show greater overlap with over-edited sites following EZH2 depletion (**Response Letter Figure 3**, also included in the revised manuscript as **Supplementary Fig. 4h**). These findings suggest that EZH2 binding tends to coincide with editing sites downregulated upon EZH2 knockdown, while ILF2 binding tends to coincide with editing sites upregulated upon EZH2 knockdown.

- the response to reviewer 1 point 4 should be conducted with the same window and

preferably a smaller one as eCLIP should give residue level information about the RNA binding footprint of each when done correctly (200nt for ADAR1 and EZH2 or 1000nt for both - it is not clear why a different window of binding would be appropriate or justified).

Response: We appreciate the reviewer's insightful comment regarding the choice of window size around eCLIP peaks. In the revised analysis, we have applied a uniform ± 200 nt window for both ADAR1 and EZH2 datasets to ensure methodological consistency. As depicted in **Response Letter Figure 4**, a significant overlap could still be observed between ADAR1-bound regions and EZH2-regulated editing sites using the same window.

-The model proposed involving all three dsRBD of ADAR1 should be supported with Alphafold predictions if possible - I find it hard to see how ADAR1 could commit all three dsRBD to interacting with EZH2 and there be both increased and decreased editing. The alternative model could be that the interaction occurs separately from editing by ADARs.

Response: We thank the reviewer for this great advice and performed Alphafold 3 predictions as suggested. As depicted in **Response Letter Figure 5**, dsRBD1 and dsRBD3 domains from ADAR1 are scored moderately well to be plausible candidates to interact physically with the SANT2 domain of EZH2, whereas dsRBD2 was scored low. Worth to note that, Alphafold 3 recognizes $ipTM \geq 0.75$ for a strong interface plausibility, $0.5-0.75$ for possible interface and < 0.5 for weak interface. We had cellular data suggesting that all three dsRBD domains from ADAR1 are critical for the interaction with EZH2 and it is possible that dsRBD1 and dsRBD3 provides direct physical interface and dsRBD2 serves as a connecting region to coordinate this interaction, so that all of them are indispensable.

Response Letter Figure 5 AlphaFold3-predicted structural models of EZH2-SANT2 domain in complex with individual ADAR1 dsRBD domains.

Protein-protein interaction predictions between EZH2-SANT2 (cyan) and ADAR1 dsRBD1 (green), dsRBD2 (yellow), or dsRBD3 (red) were generated using AlphaFold3 and visualized in PyMOL. The three panels show the top-ranked models for each domain pair. Interface predicted TM-scores (ipTM) are shown below each panel to indicate the confidence of the predicted interaction. Higher ipTM values are consistent with more confident interface predictions.

-There are multiple examples of unexpected grammar which also impair readability (e.g. line 529 - " Accordingly, it is esteemed that the landscape of ..." - esteemed is a strange choice)

Response: We thank the reviewer for pointing out this issue. We have carefully revised the manuscript to correct grammatical errors and improve readability throughout. Specifically, we have replaced “esteemed” in line 529 with “suggested”.

References:

1. Wang, X., et al., *Targeting ADAR1 with a small molecule for the treatment of prostate cancer*. *Nat Cancer*, 2025. **6**(3): p. 474-492.
2. Smoak, C.N., et al., *ZYS-1 is not an ADAR1 inhibitor*. *bioRxiv*, 2025: p. 2025.08.08.669362.
3. Zhang, M., et al., *Re: Concerns Regarding the Validation of ZYS-1 as a *Bona Fide* ADAR1 Inhibitor*. *bioRxiv*, 2025: p. 2025.03.07.641892.
4. Fritzell, K., et al., *Sensitive ADAR editing reporter in cancer cells enables high-throughput screening of small molecule libraries*. *Nucleic Acids Res*, 2019. **47**(4): p. e22.

Point-by-point response to Reviewer's comments

Dear Reviewer,

We would like to thank you again for taking time to provide additional comments on our manuscript. We have carefully considered your comments and do our best to address every one of them as follows.

Reviewer #4:

The authors have made efforts to amend the manuscript in response to the comments raised during review. It remains a very dense and complex manuscript with the chosen figure presentation not helping the reader access the important findings given how dense each figure panel is.

Response: We thank the reviewer for acknowledging our efforts in the first-round revision. In the current revision, we have made additional changes to further improve clarity and readability, including revising the presentation of figures and associated text to help readers more easily access the key findings.

Based my initial reviews and considering both the response to reviewers from the authors and the changes to the text itself I would recommend removal of the 8-Aza data. It has been demonstrated not to be an ADAR inhibitor so why the authors have decided to persist as if it is is not clear to me at all. The moderated text is the manuscript wants the reader to conclude that 8-Aza is an in cell ADAR editing regulator when this is clearly not established or supported by the data available from cell line treatments with this agent (which notably are not cited by the present authors). The current text is disingenuous when the authors acknowledge in the rebuttal that 8-Aza is not an inhibitor and then make a superficial change to the text.

Response: We agree with the reviewer on this point and have removed all 8-Aza-related data (the original **Fig. 7d and 7e**) from our manuscript.

The authors should clearly state the level of editing change considered significant for the analysis (ie what cut off was used to call a site edited and then what change (5%, 10% etc) was considered significant and how many of the replicates were required to support the changes).

Response: We thank the reviewer for requesting clarification of the criteria used to define RNA editing sites, the thresholds for significant changes, and the number of replicates analyzed. We have now explicitly stated these details in the figure legends and/or Methods section and summarized below:

For **Fig. 2b** and **Fig. 2d**, RNA editing sites were identified using REDIttoolDnaRna.py from REDIttools (v1.3) with default parameters[1]. SNPs and splicing sites were filtered out, and sites with a read coverage greater than 10 were considered as candidate editing sites. An editing level cutoff of $\geq 5\%$ was applied, and only sites with a P value < 0.05 were considered significant.

In **Fig. 2b**, the global A-to-I editing ratios were compared across groups using a two-

tailed unpaired Wilcoxon test, with n=4 biological replicates for siCTRL, n=4 for siEZH2, n=6 for siADAR1, and n=5 for siADAR2. In **Fig. 2d**, scatter plots show comparisons of individual editing sites between control and knockdown conditions in C4-2 cells using the same replicate numbers and statistical criteria.

For **Fig. 3c**, editing levels were retrieved directly from VCF files generated from the TCGA dataset, including 152 normal and 496 cancer samples.

For **Fig. 3d**, samples were stratified into EZH2-Low, EZH2-High, ADAR1-Low, and ADAR1-High (n=50 for each) groups based on expression levels.

For **Fig. 4i**, a total of n=131 genes were analyzed.

For **Fig. 6b**, analyses were performed using two independent RNA-seq biological replicates.

For **Supplementary Fig. 2c**, RNA editing sites were identified using REDIttoolDnaRna.py (REDIttools v1.3) with default cutoffs after filtering SNPs and splicing sites, retaining sites with coverage greater than 10. Each organoid sample includes two biological replicates.

For **Supplementary Fig. 2d**, due to the relatively low sequencing depth of normal poly(A) RNA-seq data (15-20 Mb per sample), two or three replicates from public datasets were merged for analysis. In this case, REDIttools de novo analysis was applied, and only editing sites with editing levels between 5% and 40% were retained, as the de novo model is more prone to false-positive calls compared with REDIttoolDnaRna.py.

For **Supplementary Fig. 2e**, n=10 samples were analyzed per group.

For **Supplementary Fig. 2f**, analyses were conducted using 4 RNA-seq biological replicates.

Why has Figure 4a have statistical analysis when it says n=2?

Response: We are sorry for this mistake and have removed the statistics in **Fig. 4a**.

The meaning of “edited/WT ratio” for the analysis of the editing in Figure 2g-2h needs to be more clearly defined. A change from 1% editing to 2% editing is a 2 fold difference but unlikely to be meaningful – the way the data has been chosen to be displayed removes the important information about the absolute level of editing (as shown in Fig 3a) of a given site from the analysis.

Response: We thank the reviewer for this important question. We agree that fold changes in the edited/WT ratio alone do not convey the absolute level of RNA editing, and that a 2-fold increase from a very low baseline may not be biologically meaningful.

In principle, RESS-qPCR is designed to measure relative changes in the abundance of edited versus wild-type transcripts between experimental conditions, rather than absolute editing frequency[2]. In this study, RESS-qPCR was primarily used to experimentally validate RNA-seq-identified editing changes. Accordingly, the edited/WT ratios shown in **Fig. 2g and 2h** are intended to reflect directional and comparative changes in editing efficiency, not absolute editing percentages.

For downstream functional characterization of a specific EZH2-mediated editing site, we employed sequencing-based approaches, including Sanger sequencing, to directly quantify absolute RNA editing levels (**Fig. 3a, 3b, and 3h**).

To address the reviewer's concern and improve clarity, we have now explicitly defined the meaning of the "edited/WT ratio" in the **Fig. 2g and 2h** figure legend as: *The edited/WT ratio reflects relative differences in editing efficiency between conditions, not absolute editing percentages.*

Line 361 – no primary reference to support editing-independent p110 functions in the cytoplasm? Cited reference is a general review.

Response: In the current manuscript, the original article which reported the editing-independent p110 functions in the cytoplasm has also been cited in Line 361[3].

Line 206-208 – no evidence genetically for compensation in vivo – this claim is not required.

Response: Following the reviewer's suggestion, we have deleted this sentence in the updated manuscript.

References

1. Picardi, E. and G. Pesole, *REDIttools: high-throughput RNA editing detection made easy*. *Bioinformatics*, 2013. **29**(14): p. 1813-4.
2. Crews, L.A., et al., *An RNA editing fingerprint of cancer stem cell reprogramming*. *J Transl Med*, 2015. **13**: p. 52.
3. Sakurai, M., et al., *ADARI controls apoptosis of stressed cells by inhibiting Staufen1-mediated mRNA decay*. *Nat Struct Mol Biol*, 2017. **24**(6): p. 534-543.